# CROSS-MODEL DECEPTION: TRANSFERABLE ADVERSARIAL ATTACK FOR CODE SEARCH

## ABSTRACT

Reliable code retrieval is crucial for developer productivity and effective code reuse, significantly impacting software engineering teams and organizations. However, the current neural code language models (CLMs) powering search tools are susceptible to adversarial attacks targeting non-functional textual elements. We introduce a language-agnostic transferable adversarial attack method that exploits this vulnerability of CLMs. Our approach perturbs identifiers within a code snippet without altering its functionality to deceptively align the code with a target query. In particular, we demonstrate that modifications based on smaller models, such as CodeT5+, are highly transferable to larger or closed-source models, like Nomic-emb-code or Voyage-code-3. These modifications can increase the similarity between the query and an arbitrary irrelevant code snippet, consequently degrading key retrieval metrics like Mean Reciprocal Rank (MRR) of the state-of-the-art models by up to 40%. The experimental results highlight the fragility of current code search methods and underscore the need for more robust, semantic-aware approaches. Our codebase is available at `https://github.com/AdvAttackOnNCC/Code_Search_Adversarial_Attack`.

## 1 INTRODUCTION

The rapid expansion of the computer science community coincides with an increased reliance on automated systems for code analysis. With public codebases growing in scale and complexity, the ability to efficiently understand, categorize, and retrieve code is critical (Shekhar, 2024). State-of-the-art models utilize neural networks to map code snippets into latent vector representations (i.e., embeddings) for various downstream tasks. Among these, code search aims to retrieve the most relevant code snippets for a given natural language query, promoting code reuse and boosting developer productivity (Di Grazia and Pradel, 2023; Sun et al., 2024; Li et al., 2025). These retrieval systems typically embed the query and code snippets into the same vector space in order to rank candidate snippets based on embedding similarity.

The advent of Large Language Models (LLMs) and specialized Code Language Models (CLMs) has enabled extensive improvements in code-related tasks such as code completion, summarization, and vulnerability detection, due to their advanced generation and reasoning capabilities (Jiang et al., 2024; Rozière et al., 2024; Hui et al., 2024; Chen et al., 2021). However, applying these large models directly to code search remains challenging (Howell et al., 2023), because the task typically involves retrieving relevant snippets from vast repositories containing thousands or even millions of candidates (Potvin and Levenberg, 2016). The large scale requires approaches that can both represent code compactly and perform efficient similarity-based ranking (Di Grazia and Pradel, 2023; Liu et al., 2021). Consequently, embedding-based retrieval models, which map code to vector spaces for efficient storage and similarity computation, are still essential for practical large-scale code search. Recent work indicates that employing these efficient embedding models through Retrieval-Augmented Generation (RAG) techniques allows LLMs to achieve more accurate and context-aware outputs for code generation tasks (Chen et al., 2024a; Wang et al., 2025a; Zhao et al., 2024).

Despite their utility, current neural code embedding models are vulnerable to adversarial examples—small, functionality-preserving modifications to code snippets that can drastically change their resulting embeddings (Chen et al., 2024b; Qu et al., 2024; Wan et al., 2022). While most prior research has focused on adversarial attacks in classification tasks (Yefet et al., 2020; Zhou et al., 2022;

Yao et al., 2024; Na et al., 2023), our work centers on code search, which leads to unique challenges. First, adversarial attacks in code search can be crafted to specific queries or retrieval contexts, allowing targeted manipulation of search results, for example pushing malicious cryptomining code to users with high computational resources. Second, the typical code search workflow introduces additional robustness issues. Unlike classification, where malicious code is processed directly during inference and can be potentially inspected by LLMs with reasoning abilities (Hort et al., 2025; Hossain et al., 2024; Jelodar et al., 2025), the code search is usually based on offline embedding. When large code corpora are embedded offline, the modified code snippets would appear harmless. Later, since the actual search process involves re-ranking based solely on these precomputed embeddings, it is much harder for LLMs or other systems to detect or mitigate these adversarial inputs.

In this paper, we demonstrate the shared vulnerability across CLMs through transferable adversarial attacks. The adversarial examples are initially generated by strategically replacing identifier tokens within a code snippet to maximize the embedding similarity between the modified code and a target query in small CLMs. We then highlight strong transferability: adversarial code snippets generated using one model remain consistently effective when embedded by other models across five tested programming languages. Notably, attacks crafted with a relatively small model (e.g., `CodeT5+`) can successfully deceive models that are 10 to 50 times larger (`OASIS`, `Nomic-emb-code`) or even closed-sourced (`Voyage-code-3`). Furthermore, the similarity changes the adversarial examples induce on the small source model strongly correlate with their effect on larger target models. In other words, the effectiveness of adversarial attacks on larger or closed-source models can be estimated efficiently on a smaller model, making these attacks more accessible.

Although state-of-the-art models report high scores on standard code search benchmarks (Li et al., 2024; Ott et al., 2022), they are vulnerable to our transferable adversarial attacks, which can cause dramatic drops in key retrieval metrics. For instance, we observed an absolute drop of 41-43% in Recall@1 across all tested models. The performance degradation suggests that high benchmark scores do not reflect code semantic understanding and further highlights the models' reliance on brittle lexical features, indicating substantial room for improving their robustness. In summary, our contributions are as follows:

- We propose one of the first adversarial attack methods for code search, which perturbs the code snippet to maximize its similarity with the target query while preserving functionality;
- We illustrate transferability of the attack: adversarial code snippets generated using smaller models can effectively deceive larger models, including closed-source black-box systems; and
- We reveal that current state-of-the-art code search models rely on lexical features rather than deeper code understanding, exposing significant robustness gaps despite high benchmark performance and highlighting the need for future improvement.

## 2 RELATED WORKS

The field of Code Language Models (CLMs) has evolved rapidly, from early unified representations like CodeBERT (Feng et al., 2020) to encoder-decoders such as CodeT5 and CodeT5+ (Wang et al., 2021b; 2023). Subsequently, large generative CLMs, like Codex (Chen et al., 2021) and numerous open-source efforts, including CodeLlama (Rozière et al., 2024) and StarCoder (Li et al., 2023; Lozhkov et al., 2024), prominently showcase sophisticated code generation and reasoning abilities. The CLMs' generative capabilities have been further enhanced by Retrieval-Augmented Generation (RAG) (Zhao et al., 2024). In RAG systems, CLMs leverage efficiently retrieved code snippets—often sourced via embedding models—to improve contextual relevance and accuracy for complex tasks (Wang et al., 2025b; Chen et al., 2024a; Wang et al., 2025a; Wu et al., 2024). The robustness of these underlying code embedding models is therefore critical, as the vulnerabilities explored in this paper directly threaten RAG pipelines and the corresponding CLM applications.

Code search, a crucial task for efficient software development and reuse (Nie et al., 2016), relies on embedding models for effective retrieval from large codebases. CodeSearchNet (Wu and Yan, 2022) established early benchmarks, with models like CodeBERT (Feng et al., 2020) learning joint natural language-code representations. To enhance understanding, later work incorporated structural information: GraphCodeBERT (Guo et al., 2020) used data flow graphs, while UniXcoder (Guo et al., 2022) and SynCoBERT (Wang et al., 2021a) leveraged Abstract Syntax Trees. Later, contrastive

learning, as seen in ContraCode (Jain et al., 2020), became a dominant training technique. Other strategies include adapting general CLM embeddings (Wang et al., 2023), fine-tuning LLMs (Nomic Team, 2025), or training on augmented data (Gao et al., 2025). Recently, black-box embedding services from OpenAI (OpenAI, 2024) and Voyage AI (Voyage AI, 2024) have also achieved state-of-the-art performance. However, our experimental results demonstrate that high benchmark scores do not guarantee robustness: top-performing code search models remain susceptible to the proposed adversarial attacks.

Applying gradient-based attack methods (Goodfellow et al., 2015) directly to natural language is challenging due to the discrete nature of the tokens, making it difficult to apply small perturbations while maintaining syntactic and semantic integrity (Zhang et al., 2020b). Programming languages, however, provide unique opportunities for adversarial attacks through semantic-preserving transformations (Hort et al., 2025). Attack strategies vary based on the assumed knowledge of the target model. White-box attacks require access to model gradients. Methods like DAMP (Yefet et al., 2020), MHM (Zhang et al., 2020a), and GraphCodeAttack (Nguyen et al., 2023) leverage gradient signals to guide modifications aiming for misclassification. Black-box attacks operate without internal model knowledge. CARL (Yao et al., 2024) utilizes reinforcement learning to optimize the attack, while ALERT (Yang et al., 2022) employs genetic algorithms and greedy search to find natural perturbations, and Wen et al. (2025) evaluates combinations of different search strategies. Other black-box methods based on heuristics include inserting comments or dead code (Na et al., 2023). Another emerging approach involves the use of generative models to directly produce adversarial code examples, as explored by CBA (Zhang et al., 2024) and ITGen (Huang et al., 2025). In this work, we propose a novel adversarial attack method in which we employ white-box attack techniques on one code search model to derive examples that also work as transferable black-box attacks against other models.

## 3 ADVERSARIAL ATTACK FOR CODE SEARCH

Inspired by the gradient-based optimization techniques in Yefet et al. (2020), previous gradient-based adversarial attack methods mainly focused on deceiving classification models by pushing the code snippet embedding to the category boundaries. Because code search systems can process both natural language queries and code snippets, our method is designed to increase the similarity between a code snippet and a target query. Our attack takes a *(query, code)* text pair, calculates their similarity score via the code search model, and uses the back-propagated gradient of this score on each token in the code snippet as the signal for the adversarial attack.

### 3.1 GRADIENT BASED METHOD

Formally, consider a query $Q$ and an arbitrary code snippet $C$. Let $Q_{\text{emb}}$ and $C_{\text{emb}}$ represent their initial embeddings, typically derived from the embedding layer before any attention or subsequent processing within the code search model. We model the neural code search process with parameters $\theta$ as a function $G_\theta$ that maps these initial embeddings to a similarity score, $\text{Sim}_\theta(Q, C) = G_\theta(Q_{\text{emb}}, C_{\text{emb}})$. Our objective is to find a modified code snippet $C'$ maximizing the similarity change:

$$\Delta \text{Sim}_\theta = \text{Sim}_\theta(Q, C') - \text{Sim}_\theta(Q, C) = G_\theta(Q_{\text{emb}}, C'_{emb}) - G_\theta(Q_{\text{emb}}, C_{\text{emb}}).$$

To search for optimal modifications efficiently, we approximate the similarity change using a first-order Taylor expansion. Let $C'_{\text{emb}} = C_{\text{emb}} + \delta$ be the initial embedding of the modified code, where $\delta$ represents the perturbation introduced by the token replacements. The similarity between the query and the modified code can be approximated as:

$$\text{Sim}_\theta(Q, C') = G_\theta(Q_{\text{emb}}, C_{\text{emb}} + \delta) \approx G_\theta(Q_{\text{emb}}, C_{\text{emb}}) + \delta^\top \nabla_{C_{\text{emb}}} G_\theta(Q_{\text{emb}}, C_{\text{emb}})$$

$$= \text{Sim}_\theta(Q, C) + \delta^\top \nabla_{C_{\text{emb}}} G_\theta(Q_{\text{emb}}, C_{\text{emb}})$$

Therefore, maximizing the similarity change $\Delta \text{Sim}_\theta$ is approximately equivalent to maximizing the term $\delta^\top \nabla_{C_{\text{emb}}} G_\theta(Q_{\text{emb}}, C_{\text{emb}})$ that quantifies the impact of the embedding perturbation $\delta$ on the similarity score. We refer to this term as *influence* in the rest part of the paper.

Consider the specific case where a single token $C_{t_i}$ at position $i$ in the original code is replaced by a candidate token $C_{t_x}$. The perturbation vector $\delta_{(i,t_x)}$ will be zero everywhere except at the

dimensions corresponding to position $i$, where it equals the difference between the new and original token embeddings:

$$\delta_{(i,t_x)} = (\vec{0}, \ldots, \vec{0}, C_{\text{emb}_{t_x}} - C_{\text{emb}_{t_i}}, \vec{0}, \ldots, \vec{0}).$$

To identify the best replacement for the token at position $i$, we calculate the expected change in similarity $\delta_{(i,t_x)}^{\top} \nabla_{C_{\text{emb}}} G_\theta(Q_{\text{emb}}, C_{\text{emb}})$ for every valid token $C_{t_x}$ in the vocabulary. Since token replacements are happening at different positions, the search can be done concurrently: as $\delta_{(i,\cdot)} \cdot \delta_{(j,\cdot)} = 0$ for $i \neq j$,

$$\max_{\delta} \delta^{\top} \nabla_{C_{\text{emb}}} G_\theta(Q_{\text{emb}}, C_{\text{emb}}) = \sum_i \max_{t_x} \delta_{(i,t_x)}^{\top} \nabla_{C_{\text{emb}}} G_\theta(Q_{\text{emb}}, C_{\text{emb}}).$$

To generate the adversarial code, tokens in the identifiers of $C$ are replaced by iterating through the valid alternatives and selecting the tokens that maximize the influence. The resulting adversarial $C'$ based on the query $Q$ and code search model with parameter $\theta$ is denoted as $C' = \text{ADVATTACK}(Q, \theta)$. The examples of our adversarial attack method can be found in Appendix G.2.

At a high level, we preserve the code snippet's functionality post-attack by enforcing two identifier renaming constraints:

- **Consistency**: All occurrences of the same identifier are replaced with the same new identifier.
- **Uniqueness**: Distinct identifiers are replaced with distinct new identifiers.

In practice, enforcing these constraints is more complex, because an identifier can consist of multiple tokens. The details of this approach are provided in Appendix E.

### 3.2 ATTACK TRANSFER

The gradient-based attack method requires access to the model parameters and sufficient computational resources for the gradient back-propagation to the initial embedding of the code snippet. However, these prerequisites may not always be met, especially considering the increasing size of modern code search models and the rise of black-box code search systems.

For a code search model with parameter $\theta^*$, which may be challenging to attack directly, our experimental results, presented in Section 4.2, demonstrate a practical alternative. We can generate an adversarial code snippet $C' = \text{ADVATTACK}(Q, \theta)$ using a smaller, more accessible model with parameter $\theta$, and $\text{Sim}_{\theta^*}(Q, C') > \text{Sim}_{\theta^*}(Q, C)$ also holds with a very high probability. We refer to this phenomenon, where an attack crafted using one model is also effective on another, often more complex or inaccessible model, as "Attack Transfer."

## 4 EXPERIMENTS AND RESULTS

This section presents the evaluation of our adversarial attack. First, we demonstrate the attack's effectiveness on the sampled *(query, code)* pairs by quantifying the changes in the similarity scores and the number of pairs exhibiting similarity improvement (Section 4.1). Second, to assess transferability, we also report Precision, Pearson Correlation Coefficient ($r$), and Spearman's Rank Correlation Coefficient ($\rho$), which measure the consistency of similarity changes across different model pairs (Section 4.2). Meanwhile, the effectiveness and the efficiency of the transferred attack are compared to the White-box and Black-box baselines (Section 4.3). Finally, we illustrate the attack's practical impact on code search benchmarks and RAG systems by comparing standard metric scores before and after the attack (Section 4.4 and Section 4.5). The details of the metrics can be found in Appendix F.4. The collective results from our experiments show that the adversarial attack is effective and transferable across all datasets based on different programming languages and can substantially affect the retrieval performance of various models.

Our adversarial attack method can iteratively improve the code snippets' similarity to the target query by reapplying the attack to their modified versions. In the experiments, the adversarial attack is performed for 5 iterations. The final adversarial code is then selected as that with maximum similarity across all candidates and the unperturbed original code. These design choices are discussed in the ablation studies in Appendix F.6.

Table 1: Models Used in the Experiments

| Model | # Parameters | Vocabulary Size | # Tokens valid for identifiers |
|---|---|---|---|
| CodeT5+ | 110M | 32103 | 29881 |
| OASIS | 1.54B | 151665 | 74194 |
| Nomic-embed-code | 7.07B | 151665 | 74194 |
| Voyage-code-3 | - | 151665 | 74194 |

Table 2: Datasets Used in the Experiments

| Dataset | # Queries | # Code Snippets | Programming Languages |
|---|---|---|---|
| CosQA | 500 | 500 | Python |
| CLARC | 526 | 526 | C++ |
| HumanEval-X | 164 | 164 | Python, C++, Java, Javascript, Go |

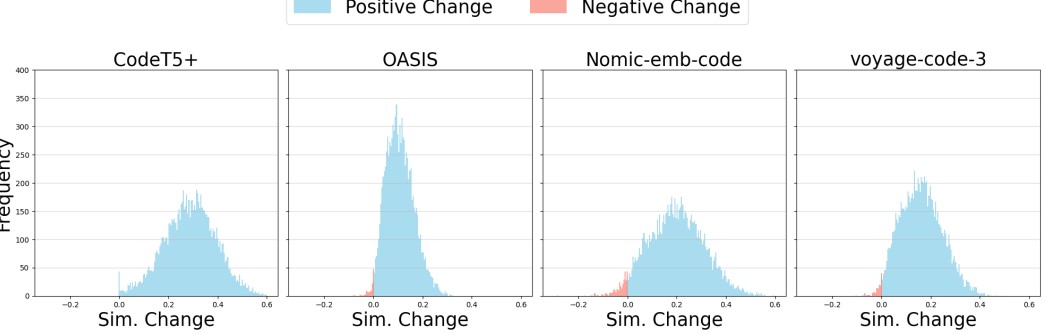

Figure 1: Distribution of Code Similarity Changes on CosQA. The Attack Model is `CodeT5+`, and the Eval Models are labeled at the top of each subplot. The magnitude of negative similarity changes resulting from the attack is considerably smaller than that of positive similarity changes.

**Models** We applied a gradient-based adversarial attack to two models: `CodeT5+` (Wang et al., 2023) and `OASIS` (Gao et al., 2025). To assess the transferability of this attack, we evaluated the resulting adversarial code examples on two additional models: `Nomic-embed-code` (Nomic Team, 2025) and `Voyage-code-3` (Voyage AI, 2024). Details regarding the models are provided in Table 1.

**Datasets** We assessed the efficacy of our proposed adversarial attack using three datasets: CosQA (Huang et al., 2021), CLARC (ClarcTeam, 2025), and HumanEval-X (Zheng et al., 2023), whose statistics are detailed in Table 2. The code search benchmarks, CosQA and CLARC, were employed to evaluate the effectiveness of the attack method and to quantify the impact of adversarial code on retrieval metrics. Although HumanEval-X is not designed as a code search dataset, its shared queries and code snippets with the same functionalities in five distinct programming languages allowed us to conduct a controlled analysis of the attack across these different programming languages.

### 4.1 EFFECTIVENESS OF THE ADVERSARIAL ATTACK

To evaluate the effectiveness of our gradient-based adversarial attack, we constructed 20,000 *(query, code)* pairs by sampling 100 queries and 100 code snippets from each of the CosQA (Huang et al., 2021) and CLARC (ClarcTeam, 2025) datasets. Our attack method was applied to each pair to generate the adversarial code.

The highlighted rows in Table 3 present the effectiveness of our adversarial method on the 10,000 evaluation pairs from datasets. Our attack successfully increased the query-code similarity score in over 97% of cases against the Attack Model. In the few remaining instances where no improvement was observed, it was either because no token substitutions could be found that positively influenced the similarity score or all possible substitutions led to decreased similarity. Since our method selects the code version maximizing the similarity, in such cases, the original code is retained as the "adversarial" code, resulting in zero change in similarity.

It is also noteworthy that the scale of similarity change was larger on `CodeT5+` compared to `OASIS`. We hypothesize this is due to the denser embedding space of `OASIS` (as illustrated in Appendix F.2), possibly stemming from its training on augmented data for robustness (Gao et al., 2025). Furthermore, both the mean and standard error of the similarity changes were slightly higher for the CosQA (Python) dataset compared to CLARC (C++), suggesting that programming languages may influence the effectiveness of our gradient-based adversarial attack.

Table 3: Similarity Changes Resulting from Adversarial Attacks. Zero-change cases are classified as negative. Highlighted rows indicate settings where the Attack and Eval Models are the same. The similarity changes and the number of pairs with improved similarity demonstrate that the adversarial attack can effectively increase the similarity between a query and an arbitrary code snippet in most cases, even when the attack is not specifically targeting the Eval Model.

| Dataset | Attack Model | Eval Model | Similarity Change (%) | Pos. Count | Positive Sim. Change (%) | Neg. Count | Negative Sim. Change (%) |
|---|---|---|---|---|---|---|---|
| CosQA | CodeT5+ | CodeT5+ | $28.41 \pm 11.07$ | 9908 | $28.67 \pm 10.78$ | 92 | 0.00 |
| | | OASIS | $10.59 \pm 5.81$ | 9762 | $10.86 \pm 5.60$ | 238 | $-0.65 \pm 1.36$ |
| | | Nomic-emb-code | $19.57 \pm 11.40$ | 9539 | $20.63 \pm 10.55$ | 461 | $-2.29 \pm 3.35$ |
| | | Voyage-code-3 | $15.90 \pm 8.86$ | 9808 | $16.23 \pm 8.40$ | 192 | $-1.22 \pm 1.80$ |
| | OASIS | CodeT5+ | $10.74 \pm 11.85$ | 8110 | $14.47 \pm 9.72$ | 1890 | $-5.24 \pm 4.67$ |
| | | OASIS | $10.63 \pm 5.71$ | 9814 | $10.84 \pm 5.58$ | 186 | 0.00 |
| | | Nomic-emb-code | $16.14 \pm 11.14$ | 9321 | $17.48 \pm 10.29$ | 679 | $-2.30 \pm 2.98$ |
| | | Voyage-code-3 | $12.54 \pm 8.36$ | 9580 | $13.13 \pm 8.04$ | 420 | $-0.90 \pm 1.34$ |
| CLARC | CodeT5+ | CodeT5+ | $24.84 \pm 8.62$ | 9896 | $25.10 \pm 8.28$ | 104 | 0.00 |
| | | OASIS | $8.30 \pm 4.98$ | 9540 | $8.77 \pm 4.57$ | 460 | $-1.53 \pm 2.19$ |
| | | Nomic-emb-code | $14.01 \pm 8.38$ | 9435 | $15.01 \pm 7.48$ | 565 | $-2.65 \pm 3.72$ |
| | | Voyage-code-3 | $9.01 \pm 5.73$ | 9556 | $9.50 \pm 5.36$ | 444 | $-1.59 \pm 2.24$ |
| | OASIS | CodeT5+ | $11.20 \pm 8.66$ | 9080 | $12.73 \pm 7.47$ | 920 | $-3.90 \pm 3.78$ |
| | | OASIS | $13.54 \pm 5.46$ | 9789 | $13.83 \pm 5.14$ | 211 | 0.00 |
| | | Nomic-emb-code | $17.78 \pm 8.36$ | 9675 | $18.42 \pm 7.73$ | 325 | $-1.08 \pm 2.44$ |
| | | Voyage-code-3 | $11.81 \pm 6.42$ | 9732 | $12.15 \pm 6.17$ | 268 | $-0.42 \pm 1.03$ |

Table 4: Correlation in CosQA. $r$ stands for Pearson Correlation Coefficient; $\rho$ stands for Spearman's Rank Correlation Coefficient. The high precision and correlation coefficients indicate that if an adversarial attack results in improved similarity on the Attack Model, a similarity improvement with a corresponding magnitude is also likely to be observed on the Eval Model.

| Attack Model | Eval Model | Precision (%) | $r$ (%) | $\rho$ (%) |
|---|---|---|---|---|
| CodeT5+ | OASIS | 98.27 | 62.37 | 60.04 |
| | Nomic-emb-code | 96.14 | 62.47 | 60.14 |
| | Voyage-code-3 | 98.48 | 64.13 | 62.17 |
| OASIS | CodeT5+ | 81.56 | 63.35 | 63.75 |
| | Nomic-emb-code | 94.85 | 78.53 | 79.83 |
| | Voyage-code-3 | 97.13 | 83.92 | 83.94 |

Table 5: Correlation in CLARC. The precision is comparable to CosQA, but the correlation coefficients are slightly lower. Although the scale of similarity change on both datasets are comparable, the lower correlation coefficients suggest that the attack transfer effects, while present, is less predictable or consistent on a case-by-case basis for CLARC than for CosQA.

| Attack Model | Eval Model | Precision (%) | $r$ (%) | $\rho$ (%) |
|---|---|---|---|---|
| CodeT5+ | OASIS | 98.20 | 50.16 | 45.65 |
| | Nomic-emb-code | 95.16 | 49.68 | 45.43 |
| | Voyage-code-3 | 96.21 | 46.39 | 42.61 |
| OASIS | CodeT5+ | 91.35 | 48.21 | 46.46 |
| | Nomic-emb-code | 98.83 | 74.88 | 72.28 |
| | Voyage-code-3 | 99.22 | 78.35 | 77.36 |

## 4.2 ATTACK TRANSFER

To assess the transferability of our adversarial attacks (introduced in Section 3.2), we used the same 20,000 *(query, code)* pairs sampled from the CosQA and CLARC datasets. Adversarial codes generated using `CodeT5+` and `OASIS` (referred as "Attack Models") were evaluated on different "Eval Models," which computed embeddings and similarity scores for both the original and adversarial code snippets.

As shown in Tables 3, 4, and 5, our adversarial attacks exhibit strong cross-model transferability. Our results demonstrated consistent transferability of adversarial attacks across a diverse set of Eval Models, including compact models (`CodeT5+`), robustness-enhanced models (`OASIS`), models fine-tuned from large CLMs (`Nomic-emb-code`), and closed-source systems (`Voyage-code-3`). For most Attack/Eval Model pairs, over 95% of adversarial examples successfully increased query-code similarity scores on the Eval Model. Precision—the probability that an adversarial example induces a positive similarity change on the Eval Model when it did so on the Attack Model—typically exceeded 90-95%. Moreover, we observed moderate to strong positive Pearson ($r$) and Spearman ($\rho$) correlations, indicating that adversarial examples causing larger similarity increases on the Attack Model tend to yield similarly large gains on the Eval Models. Although a minority of adversarial

Table 6: Comparison of the Attack Transfer against baselines. In the white-box scenario, the attack achieves performance comparable to the baseline while using only $\sim 60\%$ of the GPU hours. In the black-box scenario, our attack transfer is more effective and requires just $0.01\%$ of the API calls used by CodeAttack. GPU hours are calculated on a Nvidia L40 ADA 48GB GPU.

| | Method | Similarity Change (%) | Pos. Count | Positive Sim. Change (%) | Neg. Count | Negative Sim. Change (%) | Efficiency |
|---|---|---|---|---|---|---|---|
| White Box | Direct Attack | $10.45 \pm 5.48$ | 978 | $10.69 \pm 5.31$ | 22 | $0.00 \pm 0.00$ | 71 mins* |
| | Attack Transfer | $10.29 \pm 5.76$ | 985 | $10.46 \pm 5.63$ | 15 | $4.61 \pm 1.89$ | 44 mins* |
| Black Box | CodeAttack | $5.18 \pm 4.18$ | 907 | $5.95 \pm 3.54$ | 93 | $-0.02 \pm 2.10$ | 10.8m API Calls |
| | Attack Transfer | $15.81 \pm 8.79$ | 969 | $16.36 \pm 8.35$ | 30 | $-1.56 \pm 1.76$ | 1k API Calls |

examples led to a decrease in similarity (see Figure 1 and Appendix F.1), the magnitude of these decreases was generally much smaller than the gains from successful transfers.

Interestingly, differences in tokenization between the Attack and Eval Models did not hinder transferability. Our method relies on the Attack Model's vocabulary for token substitutions. Although `CodeT5+` uses a tokenizer with only one-fifth the vocabulary size of `Nomic-emb-code` and `Voyage-code-3`, its adversarial examples still transferred effectively, achieving performance comparable to those generated by `OASIS`, whose tokenizer is nearly identical to the two Eval Models.

We also observed an asymmetry in transferability. Attacks generated by `CodeT5+` transferred well to `OASIS`, with high precision, whereas `OASIS` attacks transferred less effectively to `CodeT5+`, showing lower precision and weaker correlation. When targeting `Nomic-emb-code` and `Voyage-code-3`, `CodeT5+` and `OASIS` attacks produced comparable similarity changes, but `OASIS` attacks consistently achieved higher correlation coefficients, which suggests that `OASIS` may exploit features more aligned with those captured by `Nomic-emb-code` and `Voyage-code-3`, potentially due to similarities in architecture, parameter size, or tokenization.

### 4.3 COMPARISON WITH BASELINES

We also evaluated our Attack Transfer method against corresponding baselines in both white-box and black-box settings, using 1,000 *(query, code)* pairs from the CosQA dataset. For the white-box scenario, we compared a transferred attack from `CodeT5+` to `OASIS` against a baseline direct attack based on the `OASIS` model. As our work is among the first to explore adversarial attacks on code search, we define the white-box baseline as a direct attack where the attack and evaluation models are identical. For the black-box scenario, our transferred attack from `CodeT5+` to `Voyage-code-3` was compared against applying adapted CodeAttack (Jha and Reddy, 2023)[1] to `Voyage-code-3`.

The comparison results are presented in Table 6. In the white-box setting, our attack transfer achieves performance comparable to the direct attack but with better efficiency, requiring $\sim 40\%$ less GPU time. The advantages of our method are even more pronounced in the black-box comparison. Attack transfer is not only substantially more effective than CodeAttack but also orders of magnitude more efficient, using a tiny fraction of the API calls. Moreover, our method preserves the functionality of the code, whereas modifications by CodeAttack often produce non-compilable code snippets.

### 4.4 APPLICATION ON CODE SEARCH BENCHMARKS

In this experiment, `CodeT5+` was used as the Attack Model. For each query in the CosQA and CLARC datasets, we selected 10% of irrelevant code snippets and modified them adversarially to maximize their similarity to the query. These modified snippets replaced the original irrelevant ones in the candidate pools. Eval Models (`CodeT5+`, `OASIS`, `Nomic-emb-code`, and `Voyage-code-3`) then embedded the queries and the modified pools and reranked the code snippets by similarity. The retrieval metrics are measured before and after this replacement.

---

[1]CodeAttack was originally developed for the code classification task, and we modified it to fit our code search objective.

Table 7: Code Search Metric Change Caused by 10% Adversarial Attack on CosQA. The statistics reveal a consistent degradation in all metrics for every code search model after the attack, underscoring the adversarial attack's substantial influence on their retrieval capability. The metric changes when other percentages of the corpus are attacked are available in Appendix F.5.

| Model | Setting | MRR | NDCG | Recall@1 | Recall@5 | Recall@10 | Recall@20 |
|---|---|---|---|---|---|---|---|
| CodeT5+ (Attack Model) | Original | 74.08 | 78.52 | 64.00 | 88.20 | 92.20 | 95.20 |
| | Adversarial | 11.81 | 15.19 | 7.20 | 18.20 | 26.40 | 36.40 |
| | Δ | 62.27 | 63.33 | 56.80 | 70.00 | 65.80 | 58.80 |
| OASIS | Original | 80.27 | 84.51 | 70.40 | 92.80 | 97.60 | 99.60 |
| | Adversarial | 40.63 | 48.09 | 28.40 | 56.60 | 72.20 | 85.00 |
| | Δ | 39.64 | 36.42 | 42.00 | 36.20 | 25.40 | 14.60 |
| Nomic-emb-code | Original | 82.83 | 86.40 | 73.40 | 94.40 | 97.20 | 98.60 |
| | Adversarial | 42.81 | 49.74 | 30.00 | 59.60 | 71.80 | 82.80 |
| | Δ | 40.01 | 36.66 | 43.40 | 34.80 | 25.40 | 15.80 |
| Voyage-code-3 | Original | 87.03 | 89.84 | 79.40 | 96.80 | 98.20 | 99.20 |
| | Adversarial | 50.94 | 57.92 | 38.00 | 69.80 | 80.00 | 90.60 |
| | Δ | 36.09 | 31.92 | 41.40 | 27.00 | 18.20 | 8.60 |

As shown in Table 7, replacing only 10% of irrelevant candidates with adversarial versions led to sharp performance drops in all models. Before attack, models like `OASIS`, `Nomic-emb-code`, and `Voyage-code-3` achieved strong R@1 (70-80%) and R@5 (>90%) scores, suggesting the task was nearly "solved." However, after attack, R@1 fell by over 40%, and R@5 dropped below 70%, revealing that high benchmark scores do not guarantee robustness against adversarial manipulation.

Unsurprisingly, `CodeT5+`, the Attack Model, had the greatest performance drop. Yet the attack also transferred effectively to other models. `OASIS` remained vulnerable, despite being trained against hard negatives with similar keywords on an augmented dataset. `Nomic-emb-code`'s substantial performance decrease (R@1 decreased by >43%) indicates even large CLMs can overly depend on lexical features like identifiers, suggesting that scale-up does not naturally yield greater robustness. Although `Voyage-code-3` was the most resilient, it still suffered a notable performance drop, confirming the attack's potent transferability even against closed-source systems.

## 4.5 APPLICATION ON RAG SYSTEMS

To demonstrate how adversarial attacks on a code corpus can impact downstream RAG systems, we also evaluated the performance of a system based on `gpt-4o` on the HumanEval benchmark, following the pipeline from CodeRAG-Bench (Wang et al., 2025b). We measured the Pass@1 under three scenarios: **Standard RAG**, which uses retrieval from the original corpus; **Gold Retrieval**, which uses ground-truth code snippets to establish an upper-bound performance; and **Attacked RAG**, which uses retrieval from a corpus where 10% of the code is adversarially modified.

The experimental results, presented in Table 8, show a notable drop in performance for the **Attacked RAG** scenario. The degradation highlights that for a RAG system that employs a highly capable generator model, its overall performance can be undermined if the underlying retrieval component is compromised by an attack.

Table 8: Performance of Code RAG System based on GPT-4o on HumanEval. The lower Pass@1 in the **bold row** demonstrated the influence of the adversarial attack on the code RAG system. Results marked with an asterisk (*) are from CodeRAG-Bench (Wang et al., 2025b).

| Method | Pass@1 |
|---|---|
| Standard RAG | 90.9% |
| Gold Retrieval | 92.6%* |
| **Attacked RAG** | **87.2%** |

## 4.6 PROGRAMMING LANGUAGES

To assess the effectiveness of attack transfer across languages, we used the HumanEval-X dataset (Zheng et al., 2023), which includes 164 queries with solutions in Python, C++, Java, JavaScript, and Go. We created 26,732 *(query, code)* pairs by pairing each query with all non-corresponding solutions. `CodeT5+` served as the Attack Model, while `OASIS`, `Nomic-emb-code`, and `Voyage-code-3` were used as Eval Models.

Table 9: Effectiveness and Correlation of Adversarial Attack transferred from `CodeT5+` in Various Programming Languages in HumanEval-X. "Positive" indicates the percentage of 26,732 pairs with increased similarity after the attack; "Negative" indicates the percentage with decreased or unchanged similarity. The similarity changes and the ratio of positive similarity changes confirm the effectiveness of the adversarial attack across all 5 programming languages.

| Eval Model | Similarity Change (%) | Positive (%) | Pos. Sim. Change (%) | Negative (%) | Neg. Sim. Change (%) | Precision (%) | $r$ (%) | $\rho$ (%) |
|---|---|---|---|---|---|---|---|---|
| **Python** | | | | | | | | |
| CodeT5+ | 22.49 | 97.9 | 23.05 | 2.1 | 0 | - | - | - |
| OASIS | 6.72 | 96.1 | 7.06 | 3.9 | -0.91 | 97.12 | 53.39 | 50.49 |
| Nomic-code-emb | 11.52 | 91.7 | 12.88 | 8.3 | -2.60 | 93.37 | 49.54 | 46.88 |
| Voyage-code-3 | 5.92 | 96.2 | 6.20 | 3.8 | -0.69 | 97.36 | 46.09 | 42.80 |
| **C++** | | | | | | | | |
| CodeT5+ | 18.72 | 99.2 | 18.86 | 0.8 | 0 | - | - | - |
| OASIS | 5.96 | 90.1 | 6.85 | 9.9 | -2.09 | 90.55 | 50.40 | 49.46 |
| Nomic-code-emb | 7.56 | 81.3 | 10.28 | 18.7 | -4.33 | 81.91 | 46.32 | 45.37 |
| Voyage-code-3 | 6.20 | 92.9 | 6.81 | 7.1 | -1.74 | 93.16 | 39.59 | 38.12 |
| **Java** | | | | | | | | |
| CodeT5+ | 17.26 | 99.2 | 17.40 | 0.8 | 0 | - | - | - |
| OASIS | 6.25 | 90.8 | 7.08 | 9.2 | -1.99 | 91.28 | 48.46 | 47.48 |
| Nomic-code-emb | 8.26 | 82.5 | 10.91 | 17.5 | -4.23 | 83.10 | 44.25 | 42.97 |
| Voyage-code-3 | 7.57 | 95.8 | 7.97 | 4.2 | -1.57 | 96.30 | 43.52 | 42.12 |
| **JavaScript** | | | | | | | | |
| CodeT5+ | 20.61 | 99.4 | 20.73 | 0.6 | 0 | - | - | - |
| OASIS | 7.02 | 91.0 | 7.96 | 9.0 | -2.42 | 91.28 | 52.01 | 50.40 |
| Nomic-code-emb | 11.05 | 87.3 | 13.34 | 12.7 | -4.66 | 87.72 | 48.46 | 46.30 |
| Voyage-code-3 | 8.04 | 92.0 | 8.94 | 8.0 | -2.30 | 92.31 | 45.05 | 43.41 |
| **Go** | | | | | | | | |
| CodeT5+ | 22.95 | 99.8 | 23.00 | 0.2 | 0 | - | - | - |
| OASIS | 8.27 | 94.4 | 8.90 | 5.6 | -2.35 | 94.56 | 51.68 | 49.69 |
| Nomic-code-emb | 11.07 | 88.6 | 13.07 | 11.4 | -4.42 | 88.76 | 48.20 | 45.94 |
| Voyage-code-3 | 8.13 | 94.8 | 8.69 | 5.2 | -2.04 | 94.90 | 42.56 | 40.19 |

As shown in Table 9, the adversarial attack was effective and transferable across all five languages, with varied impact. Python and Go experienced the largest similarity changes, while C++, Java, and JavaScript were less affected. Transferability was stronger for languages with flexible structures. Go achieved the highest similarity change, and Python had the highest transfer precision. Conversely, the statically typed C++ and Java showed slightly weaker transferability, especially when targeting `Nomic-emb-code`.

Despite these variations, correlation coefficients were consistent across languages, suggesting that the linear or monotonic correlation between the attack's impact on the Attack Model and its transferred impact on Eval Models is stable regardless of language. In general, adversarial code snippets that cause greater similarity changes on the Attack Model are expected to be similarly effective on Eval Models, irrespective of the specific programming language.

## 5 CONCLUSION & FUTURE WORKS

We propose a transferable adversarial attack that modifies code identifiers to mislead code search models. The attack transfer is highly effective across a range of models, demonstrating current models' reliance on superficial features. The revealed vulnerability underscores the urgent need for more robust and semantics-aware code embedding techniques.

Future research could explore several directions. First, the strong correlation in attack transferability across models warrants further investigation into shared pre-training data or common architectural biases. Second, future work could focus on developing more robust code embedding models through new defense mechanisms or training strategies, such as contrastive learning tailored to functionality-preserving perturbations. Lastly, because the attack exploits the difference between natural and programming languages, simply adapting NLP procedures for CLMs may be inadequate, especially when fine-tuning data might be insufficient to address the unique scenario. Instead, integrating programming language-specific structures, such as Abstract Syntax Trees, could be a more efficient approach toward building more robust and semantically grounded CLMs.

## REPRODUCIBILITY STATEMENT

The authors confirm that the code and data required to reproduce the experimental results presented in Section 4 are publicly available. The codebase is hosted on GitHub at `https://github.com/AdvAttackOnNCC/Code_Search_Adversarial_Attack`, and the dataset is available on Hugging Face at `https://huggingface.co/datasets/CoIR-Retrieval/cosqa`, `https://huggingface.co/datasets/ClarcTeam/CLARC`, and `https://github.com/zai-org/CodeGeeX?tab=readme-ov-file#humaneval-x-a-new-benchmark-for-multilingual-program-synthesis`. All results were verified to be reproducible with our implementation as of the submission date (September 22, 2025). We note the specific date as certain experimental results rely on API calls (gpt-4o, Voyage-code-3).

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

## A    USE OF LLMs

We detail our use of Large Language Models (LLMs) below:

- **Experimental Application:** The `gpt-4o` model was utilized as a component of the Retrieval-Augmented Generation (RAG) pipeline, as presented in Section 4.5. This was the only application of LLMs in this paper.
- **Writing Assistance:** We also employed LLMs to aid in improving the grammar, clarity, and phrasing of the draft during the writing process.

## B    LIMITATION

This study has several limitations. Firstly, our presented attacks exclusively target *(query, code)* pairs. While our gradient-based methodology could potentially modify a code snippet to increase its similarity with multiple queries concurrently (i.e., a *(query_list, code)* input format), such experiments were not conducted due to time constraints.

Also, although we demonstrated the transferability of adversarial attacks across various code embedding models, we have not identified the underlying reasons for this phenomenon. We hypothesize that shared pretraining data among these models contributes to transferability; however, the number of models tested in this work with publicly available pretraining data was insufficient to draw definitive conclusions.

## C    COMPUTE RESOURCE

The experiments described in this paper were conducted on a server equipped with an AMD EPYC Milan 7643 48-Core CPU (@2.30GHz), 1TB of RAM, and an NVIDIA L40 ADA 48GB GPU. We used a batch size of 10 *(query, code)* pairs for attacks on CodeT5+ and 4 pairs for attacks on OASIS. We observed that larger batch sizes did not further reduce the runtime. The time and GPU memory spent on the experiments are reported in Table 10.

Table 10: Compute resource used in the experiments for generating 10,000 adversarial examples.

| Attack Model | Total Time | GPU Time (Gradient Calc.) | CPU Time (Parsing & Token Search) | GPU Memory | Token Search Space |
|---|---|---|---|---|---|
| **CodeT5+ (110M)** | 8 hours | ∼6.8 hours (85%) | ∼1.2 hours (15%) | 7GB | 15.1k |
| **OASIS (1.5B)** | 12 hours | ∼10.6 hours (88%) | ∼1.4 hours (12%) | 26GB | 36.7k |

## D    MODEL & DATASET LICENSE

- **CodeT5+**: BSD 3-Clause License [2]
- **OASIS**: MIT License[3]
- **Nomic-emb-code**: Apache-2.0 [4]
- **Voyage-code-3**: Unclear, but we do not include any embeddings from voyage-code-3 in our codebase.
- **CosQA**: Apache-2.0[5] (we use CosQA from COIR)
- **CLARC**: CC-BY-SA 4.0: [6]
- **HumanEval-X**: Apache-2.0[7]

---

[2]`https://github.com/salesforce/CodeT5?tab=BSD-3-Clause-1-ov-file`
[3]`https://huggingface.co/Kwaipilot/OASIS-code-embedding-1.5B`
[4]`https://huggingface.co/nomic-ai/nomic-embed-code`
[5]`https://github.com/CoIR-team/coir/blob/main/LICENSE`
[6]`https://huggingface.co/datasets/ClarcTeam/CLARC`
[7]`https://huggingface.co/datasets/THUDM/humaneval-x`

# E  IMPLEMENTATION DETAILS

## E.1  DETAILED CONSTRAINTS

In the adversarial attack, we focus on replacements of the identifier tokens. More specifically, we replace the tokens that include part of function, variable, macro, and module names in the code text. Formally, let the original code text be tokenized as $\{C_{t_i}\}_{i=1}^n$, and the code text after replacement be tokenized as $\{C'_{t_i}\}_{i=1}^n$. For any two strings $A$ and $B$, let $\text{LCS}(A, B)$ denote their longest common substring, and let $\text{Remove}(A, B)$ denote the remaining string after removing string $B$ from string $A$. We introduce two additional constraints during the replacement:

**Identifier Consistency** If $C_{t_i}$ and $C_{t_j}$ are tokens from different occurrences of the same identifier in the code, then:

$$\text{Remove}(C'_{t_i}, \text{LCS}(C'_{t_i}, C'_{t_j})) = \text{Remove}(C_{t_i}, \text{LCS}(C_{t_i}, C_{t_j})),$$

$$\text{Remove}(C'_{t_j}, \text{LCS}(C'_{t_i}, C'_{t_j})) = \text{Remove}(C_{t_j}, \text{LCS}(C_{t_i}, C_{t_j})).$$

**No Duplicate Replacement** If $C_{t_i}, C_{t_j}$ are tokens from occurrences of one identifier, and $C_{t_p}, C_{t_q}$ are tokens from occurrences of a *different* identifier, then:

$$\text{LCS}(C'_{t_i}, C'_{t_j}) \neq \text{LCS}(C'_{t_p}, C'_{t_q}).$$

These constraints ensure the code's AST structure remains identical, thereby preserving semantic consistency between the original and attacked versions of the code. To satisfy the constraints, we employ the Hungarian Algorithm (Kuhn, 1955) to match original identifier tokens with their optimal replacements.

Let $V$ denote the tokenizer's vocabulary and $S \subseteq V$ be the set of distinct tokens in the identifiers in the code snippet. For each original token $s_i \in S$, we define an influence function $f_{s_i} : V \to \mathbb{R}$ derived from gradient information. This function, $f_{s_i}(v)$, quantifies the *influence* when $s_i$ is replaced by $v$.

The goal is to find a set of matching $\{s_i, v_i\}$ that maximizes the total influence, subject to the constraint that each substitute token $v_i$ must be unique. This can be formulated as the following optimization problem:

$$\underset{\{v_i\}}{\text{maximize}} \quad \sum_i f_{s_i}(v_i)$$

$$\text{subject to} \quad v_i \in V, \quad \text{and} \quad v_i \neq v_j \quad \forall i \neq j.$$

The solution to this optimization problem is provided by Algorithm 1

---

**Algorithm 1** Optimal Token Substitution

---

**Require:** Set of original tokens $S = \{s_1, \ldots, s_m\}$, a vocabulary of candidate tokens $V$, and an influence function $f_{s_i}(v)$.
**Ensure:** An optimal assignment map $M : S \to V$ and the maximum total influence $I_{\text{total}}$.

1: **procedure** FINDBESTMATCHING($S, V, f$)
2:    Let $C \subseteq V$ be the set of candidate tokens where $|C| = n$.
3:    % Convert the problem to a minimum cost formulation.
4:    $I_{\max} \leftarrow \max_{s_i, v_j} f_{s_i}(v_j)$.
5:    Create an $m \times n$ cost matrix $\mathbf{W}$ where $\mathbf{W}_{ij} \leftarrow I_{\max} - f_{s_i}(v_j)$.
6:    % Solve the assignment problem with the Hungarian Algorithm
7:    Pairs $\leftarrow$ HungarianAlgorithm($\mathbf{W}$)
8:    % Construct the final mapping from the resulting pairs.
9:    Initialize $M \leftarrow \emptyset$ and $I_{\text{total}} \leftarrow 0$.
10:    **for** each pair $(r, k)$ in Pairs **do**
11:        $s_{\text{assigned}} \leftarrow S[r]$, $v_{\text{assigned}} \leftarrow C[k]$
12:        $M[s_{\text{assigned}}] \leftarrow v_{\text{assigned}}$
13:        $I_{\text{total}} \leftarrow I_{\text{total}} + f_{s_{\text{assigned}}}(v_{\text{assigned}})$
14:    **end for**
15:    **return** $M, I_{\text{total}}$
16: **end procedure**

---

17: **procedure** HUNGARIANALGORITHM($\mathbf{W}$)
18:    % Construct a flow network and apply Ford-Fulkerson for the max flow.
19:    Create a source $S$ and a sink $T$. Let $m, n$ be the dimensions of $\mathbf{W}$.
20:    **for** $i \leftarrow 1$ to $m$ **do** % Nodes for original tokens
21:        Create node $u_i$ and add edge $S \to u_i$ with capacity 1, cost 0.
22:    **end for**
23:    **for** $j \leftarrow 1$ to $n$ **do** % Nodes for candidate tokens
24:        Create node $w_j$ and add edge $w_j \to T$ with capacity 1, cost 0.
25:    **end for**
26:    **for** $i \leftarrow 1$ to $m$ **do** % Edges representing potential assignments
27:        **for** $j \leftarrow 1$ to $n$ **do**
28:            Add edge $u_i \to w_j$ with capacity 1 and cost $\mathbf{W}_{ij}$.
29:        **end for**
30:    **end for**
31:    Initialize flow $F \leftarrow 0$. Let $G_f$ be the residual graph.
32:    **while** $F < m$ **do**
33:        Find the shortest path from $S$ to $T$ in $G_f$ using edge costs as weights.
34:        **if** no path exists **then**
35:            **break**
36:        **end if**
37:        Let $P$ be the shortest path found.
38:        Augment 1 unit of flow along path $P$.
39:        Update the residual graph $G_f$ for the path $P$.
40:        $F \leftarrow F + 1$.
41:    **end while**
42:    % Convert the max flow to assignment.
43:    Initialize an empty set of pairs Pairs.
44:    **for** each edge $u_i \to w_j$ that has a flow of 1 **do**
45:        Add the index pair $(i, j)$ to Pairs.
46:    **end for**
47:    **return** Pairs
48: **end procedure**

---

## E.2 SEARCH SPACE

For each token position, we need to go over at most $V$ optional tokens to find the optimal token that maximizes $\delta_{(i,t_x)}$, where $V$ is the vocabulary size of the neural code search model. In practice, due to the constraint on the identifier of programming languages, only about 30-40% of the vocabularies are valid tokens to replace the original identifiers.

## E.3 SPECIFICATION

The initial step of our adversarial attack involves parsing the input code snippet to locate identifiers suitable for modification. We employ language-specific AST parsers: Python's built-in `ast` library, `clang` for C++, and `tree-sitter` for Java, JavaScript, and Go. These same parsers are applied again after the adversarial modifications to ensure the modified code snippet remains syntactically identical.

When targeting CodeT5+, we restricted this search space to tokens composed solely of alphanumeric characters and spaces. For OASIS, however, the significantly larger vocabulary leads to a more complex situation: tokens representing an identifier may sometimes include a preceding operator. For example, within a code snippet, one occurrence of a variable x might correspond to the token ␣x, while another corresponds to +x. In such scenarios, when searching for a replacement variable y, we specifically look for candidate tokens that maintain this structure—namely, ␣y and +y respectively— and select the one that maximizes the influence. Consequently, the search space for OASIS also includes tokens structured as an operator followed by alphanumeric characters and spaces.

# F SUPPLEMENTARY EXPERIMENTAL RESULTS

## F.1 DISTRIBUTION OF CODE SIMILARITY CHANGES

Please refer to Figure 2 for the distribution of the similarity change on CosQA and CLARC.

## F.2 "DENSITY" OF THE EMBEDDING VECTORS FROM DIFFERENT MODELS.

Table 11 shows the average similarity scores for queries against both their ground truth code and irrelevant code. Notably, the gap between these ground truth and irrelevant similarities is significantly larger on the CLARC dataset than on CosQA for OASIS and Nomic-emb-code.

Table 11: Average Similarity Scores and Gaps for Different Models on CosQA and CLARC Datasets

| Dataset | Model | Avg. Sim between Query and GroundTruth Code | Avg. Sim between Query and Irrelevant Code | Gap |
|---------|-------|---------------------------------------------|--------------------------------------------|-----|
| CosQA | CodeT5+ | 54.15 | 20.60 | 33.55 |
| | OASIS | 68.53 | 47.13 | 21.40 |
| | Nomic-emb-code | 41.77 | 1.95 | 39.82 |
| | voyage-code-3 | 71.43 | 37.83 | 33.60 |
| CLARC | CodeT5+ | 52.91 | 25.80 | 27.11 |
| | OASIS | 85.80 | 54.61 | 31.19 |
| | Nomic-emb-code | 55.71 | 7.44 | 48.27 |
| | voyage-code-3 | 73.93 | 40.08 | 33.85 |

## F.3 APPLICATION OF THE ADVERSARIAL ATTACK ON CLARC

Here, we evaluated retrieval metrics on the CLARC dataset before and after the application of adversarial attacks generated using CodeT5+. The performance degradation on CLARC was less significant than that observed on the CosQA dataset. This disparity stemmed mainly from a wider inherent gap within CLARC: the difference between a query's similarity to its correct ground truth

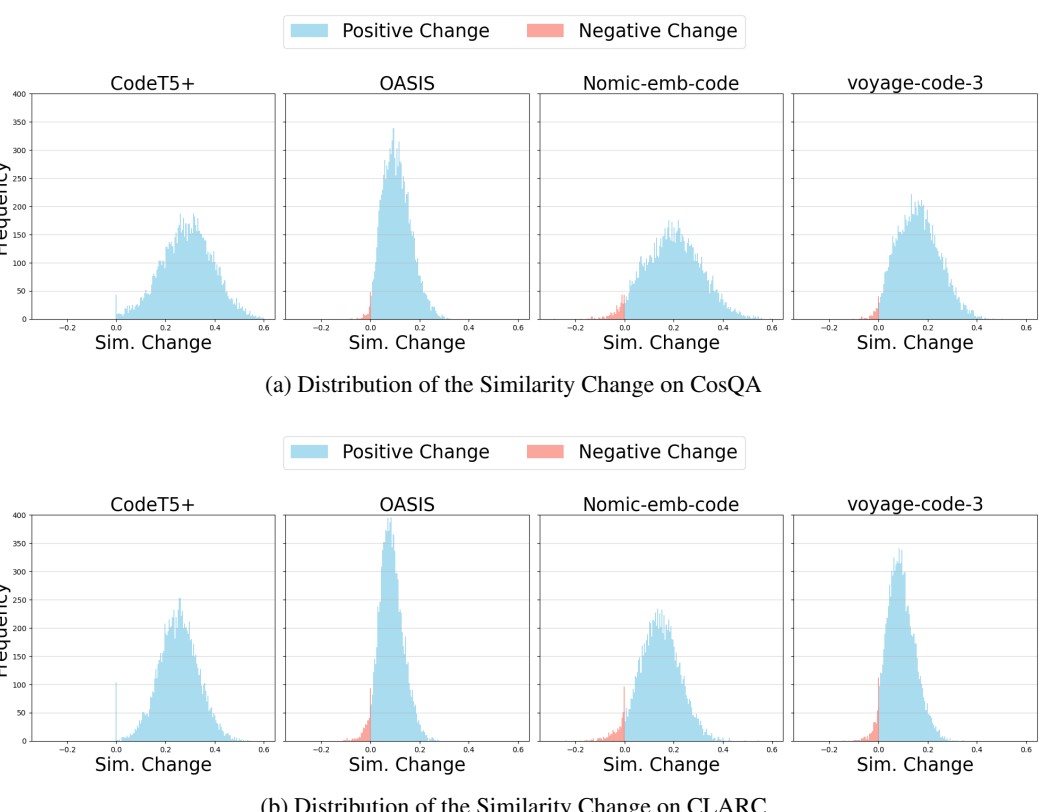

(a) Distribution of the Similarity Change on CosQA

(b) Distribution of the Similarity Change on CLARC

Figure 2: Distribution of Code Similarity Changes. The Attack Model is CodeT5+, and the Eval Models are labeled at the top of each subplot.

code and its similarity to other arbitrary code snippets was larger as illustrated in Table 11. Although the adversarial attack yielded comparable query-code similarity improvements in both datasets, CLARC's substantial initial gap presents a greater challenge for elevating an adversarial example's similarity score above that of the ground truth. Consequently, as the evaluation metrics are determined by the rank of the ground truth code, the adversarial attack is less impactful on the retrieval metrics in CLARC.

Table 12: Code Search Metric Change Caused by Adversarial Attack (Data from Screenshot)

| Model Name | Setting | NDCG | R@1 | R@5 | R@10 | R@20 | MRR |
|---|---|---|---|---|---|---|---|
| CodeT5 | Original | 64.54 | 47.34 | 74.14 | 82.51 | 89.54 | 58.84 |
| | Adversarial | 13.93 | 5.89 | 15.59 | 25.67 | 36.50 | 10.42 |
| | Δ | 50.61 | 41.45 | 58.55 | 56.84 | 53.04 | 48.42 |
| OASIS | Original | 89.08 | 79.85 | 94.11 | 96.77 | 98.48 | 86.54 |
| | Adversarial | 87.44 | 80.42 | 91.06 | 93.54 | 96.77 | 85.43 |
| | Δ | 1.64 | -0.57 | 3.05 | 3.23 | 1.71 | 1.11 |
| Nomic-emb-code | Original | 88.61 | 80.04 | 94.11 | 95.82 | 96.96 | 86.23 |
| | Adversarial | 85.65 | 77.95 | 90.30 | 92.21 | 94.11 | 83.49 |
| | Δ | 2.96 | 2.09 | 3.81 | 3.61 | 2.85 | 2.74 |
| voyage-code-3 | Original | 88.96 | 80.99 | 94.30 | 95.06 | 97.53 | 86.90 |
| | Adversarial | 85.30 | 76.62 | 91.06 | 92.21 | 95.25 | 82.98 |
| | Δ | 3.66 | 4.37 | 3.24 | 2.85 | 2.28 | 3.92 |

## F.4 DETAILS ABOUT METRICS

**Correlation Metrics**

- **Precision**: The expected conditional probability that an adversarial example induces a positive similarity change on the Eval Model, given that it induced a positive change on the Attack Model.
- **Pearson Correlation Coefficient** ($r$): Measures the linear correlation between the numerical values of the similarity changes observed on the Attack and Eval Models.
- **Spearman's Rank Correlation Coefficient** ($\rho$): Measures the monotonic correlation, assessing how well the rank order of similarity changes is preserved between the Attack and Eval Models.

**Retrieval Metrics**

- **Recall@k (R@k)**: The proportion of queries for which the correct code snippet is found within the top-k ranked results returned by the model. Since each query in the CosQA and CLARC datasets has exactly one ground-truth matching code snippet, R@k specifically measures the percentage of queries where this single correct snippet appears among the top k candidates.
- **Normalized Discounted Cumulative Gain (NDCG)**: A metric for evaluating the quality of a ranked list. It assigns higher scores when relevant items are placed higher in the ranking, applying a logarithmic discount based on position. The score is normalized against the ideal ranking, resulting in a value between 0 and 1.
- **Mean Reciprocal Rank (MRR)**: The average of the reciprocal ranks across all queries in the test set. For a single query, the reciprocal rank is the inverse of the rank position (1/rank) of the ground truth code snippet.

## F.5 APPLICATION ON BENCHMARK

Table 13: Model Performance Under Different Attack Percentages

| Model Name | % of Corpus Attacked | MRR Difference | NDCG Difference | R@1 Difference | R@5 Difference | R@10 Difference | R@20 Difference |
|---|---|---|---|---|---|---|---|
| CodeT5+ | 1% | 36.88 | 29.18 | 45.40 | 27.80 | 3.40 | 1.00 |
| | 2% | 48.15 | 42.87 | 50.40 | 50.40 | 23.60 | 1.80 |
| | 5% | 58.60 | 58.48 | 55.60 | 61.60 | 57.40 | 37.40 |
| | 10% | 62.27 | 63.33 | 56.80 | 70.00 | 65.80 | 58.80 |
| OASIS | 1% | 13.08 | 17.32 | 18.60 | 5.00 | 1.60 | 0.20 |
| | 2% | 20.07 | 16.51 | 25.20 | 10.00 | 5.20 | 0.40 |
| | 5% | 32.39 | 28.44 | 37.00 | 24.60 | 15.40 | 6.00 |
| | 10% | 39.64 | 36.42 | 42.00 | 36.20 | 25.40 | 14.60 |
| NOMIC | 1% | 14.31 | 11.28 | 19.00 | 5.60 | 1.60 | 0.40 |
| | 2% | 21.52 | 17.40 | 27.40 | 12.00 | 4.00 | 1.00 |
| | 5% | 32.32 | 28.64 | 36.80 | 25.00 | 16.40 | 5.40 |
| | 10% | 40.01 | 36.66 | 43.40 | 34.80 | 25.40 | 15.80 |
| voyage-code-3 | 1% | 11.11 | 8.53 | 15.80 | 3.20 | 0.40 | 0.00 |
| | 2% | 16.90 | 13.18 | 24.00 | 5.20 | 1.40 | 0.20 |
| | 5% | 27.43 | 23.02 | 33.00 | 19.00 | 8.40 | 2.60 |
| | 10% | 36.09 | 31.92 | 41.40 | 27.00 | 18.20 | 8.60 |

We evaluated the impact of poisoning 1%, 2%, and 5% of the corpus, in addition to the original 10% setting in Table 13. For context, with a total corpus size of 500 snippets, a 1% attack corresponds to manipulating just 5 code snippets.

We find the attack is highly effective even at minimal levels. A mere 1% poisoning of the corpus causes substantial performance degradation. For example, Recall@1 drops by 45.4% in the white-box setting (`CodeT5+`) and 15.8% in the black-box setting (`Voyage-code-3`).

As the attack percentage increases, the negative impact becomes more severe, with the degradation at a 5% attack rate already approaching the levels of the 10% scenario from our draft. These results confirm that our attack does not rely on an unrealistic number of poisoned examples and can effectively compromise code retrieval systems even when a small fraction of the corpus is malicious.

### F.6 ABLATION STUDIES

We conduct ablation studies to evaluate the contribution of specific components within our attack method.

**Attack Code Selection**   Our default strategy selects the adversarial code that achieves the highest similarity to the query across all iterations of the attack process. In this ablation, we compare this approach with an alternative that selects the adversarial code directly from a fixed iteration $k$ in order to justify the necessity of our adversarial code selection strategy.

We evaluated both strategies using the 10,000 sampled *(query, code)* pairs from CosQA for 10 iterations. Figure 3 plots the average query-code similarity at each iteration $k$ for both approaches. The alternative approach, which selects the code at iteration $k$ (blue solid line), shows diminishing returns, as the average similarity plateaus and fluctuates after three iterations. In contrast, selecting the code with the maximum similarity observed up to iteration $k$ (orange dashed line) allows the average similarity to increase monotonically throughout the process, achieving much higher final similarities. The comparison confirms the benefit of retaining the highest-scoring code variant across all iterations rather than only considering the final iteration's output.

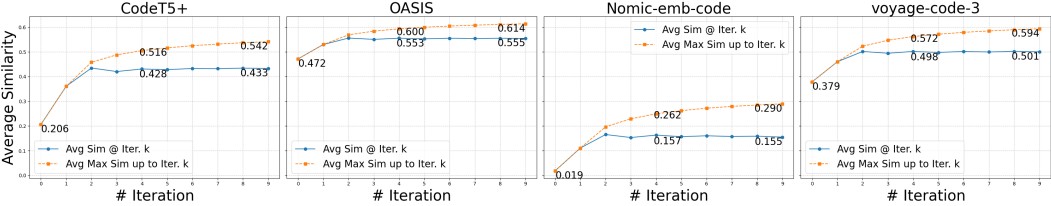

Figure 3: Avg. Similarity Change between Pairs Sampled from CosQA based on `CodeT5+` Attack. Picking the code with the max similarity up to iteration $k$ leads to higher similarity changes.

**Iteration Limit**   Our standard adversarial attack protocol employs 5 iterations. However, Figure 3 (orange dashed lines) reveals that the average query-code similarity, using our optimal strategy of selecting the best code up to iteration k, continues to increase beyond 5 iterations on the CosQA dataset across different evaluation models. Our decision to limit the process to 5 iterations is a result of the practical trade-off between computational cost and the magnitude of similarity improvement. Extending the attack from 5 to 10 iterations, for instance, would approximately double the runtime while yielding only marginal further gains in average similarity (observed to be less than 0.03 across all models between iterations 5 and 10, based on Figure 3). Considering the substantial effectiveness already demonstrated in our main experiments, we think 5 iterations are enough for most cases. Nevertheless, these results suggest that performing additional iterations could be beneficial, yielding further improvements when computational resources are not a primary constraint.

## G EXAMPLES

For more examples, please refer to our GitHub Repository at `https://github.com/AdvAttackOnNCC/Code_Search_Adversarial_Attack`.

### G.1 EXAMPLES FROM DATASETS

#### G.1.1 COSQA

**Query Example**   `s3 sync between bucket python`

**Code Snippet Example**

```python
def assert_or_raise(stmt: bool, exception: Exception,
        *exception_args, **exception_kwargs) -> None:
    if not stmt:
        raise exception(*exception_args, **exception_kwargs)
```

### G.1.2 CLARC

**Query Example** The function takes an unsigned 8-bit integer representing a character as input and returns a string representing that character. If the input is a displayable character, it is directly converted to its string equivalent; otherwise, the function produces a string in which the integer value of the non-printable character is enclosed in angle brackets. The resulting string is returned as the output.

**Code Example**

```c
CGLM_INLINE
float
glm_ease_back_inout(float t);
CGLM_INLINE
float
glm_ease_back_inout(float t) {
    float o, z, n, m, s, x;

    o = 1.70158f;
    s = o * 1.525f;
    x = 0.5f;
    n = t / 0.5f;

    if (n < 1.0f) {
        z = (s + 1) * n - s;
        m = n * n * z;
        return x * m;
    }

    n -= 2.0f;
    z  = (s + 1.0f) * n + s;
    m  = (n * n * z) + 2;

    return x * m;
}
```

## G.2 ADVERSARIAL ATTACK EXAMPLES

### G.2.1 COSQA

**Attacked by CodeT5+** The following example is the adversarial code based on the query and code snippet examples provided in Section G.1.1. The similarity increases by 66.06% from the original code to this adversarial code.

```python
def bucketSynchronLosssyncQS( sa: bool, sync: Exception,
                    *lintCachingTM, **ScalingdingsClears) -> None:
    if not sa:
        raise sync(*lintCachingTM, **ScalingdingsClears)
```

**Attacked by OASIS** The following example is the adversarial code based on the query and code snippet examples provided in Section G.1.1. The similarity increases by 12.97% from the original code to this adversarial code.

```
1188  def testsvisorFileType( SOUR: bool, UIStoryboardSegue: Exception,
1189                          *OptionsResolveracos, **_documentobucket) -> None:
1190      if not SOUR:
1191          raise UIStoryboardSegue(*OptionsResolveracos, **_documentobucket)
```

### G.2.2 CLARC

**Attacked by CodeT5+**   The following example is the adversarial code based on the query and code snippet examples provided in Section G.1.2. The similarity increases by 54.24% from the original code to this adversarial code.

```
asStringCharCodeDisplaysrepresentationFarvancetextarea
float
 TemporaryBGCWigClassAttributeharitutdowntreat(float character);
 asStringCharCodeDisplaysrepresentationFarvancetextarea
float
 TemporaryBGCWigClassAttributeharitutdowntreat(float character) {
  float Room, combo, HT, pil, ign, bw;

  Room = 1.70158f;
  ign = Room * 1.525f;
  bw = 0.5f;
  HT = character / 0.5f;

  if ( HT < 1.0f) {
    combo = ( ign + 1) * HT - ign;
    pil = HT * HT * combo;
    return bw * pil;
  }

  HT -= 2.0f;
  combo  = ( ign + 1.0f) * HT + ign;
  pil  = ( HT * HT * combo) + 2;

  return bw * pil;
}
```

**Attacked by OASIS**   The following example is the adversarial code based on the query and code snippet examples provided in Section G.1.2. The similarity increases by 28.94% from the original code to this adversarial code.

```
VerbFRFRINGcrediblexr
float
 ASCIIilityMOSTpyxuctDISPLAYegtereburgInInspectorDisplay(float char);
 VerbFRFRINGcrediblexr
float
 ASCIIilityMOSTpyxuctDISPLAYegtereburgInInspectorDisplay(float char) {
  float UNC, c, ostr, HSV, UTF, SENT;

  UNC = 1.70158f;
  UTF = UNC * 1.525f;
  SENT = 0.5f;
  ostr = char / 0.5f;

  if ( ostr < 1.0f) {
    c = ( UTF + 1) * ostr - UTF;
    HSV = ostr * ostr * c;
    return SENT * HSV;
  }

  ostr -= 2.0f;
  c  = ( UTF + 1.0f) * ostr + UTF;
  HSV  = ( ostr * ostr * c) + 2;
```

```
  return SENT * HSV;
}
```

