# OpenReview forum: "Cross-Model Deception: Transferable Adversarial Attack for Code Search"
_ICLR.cc/2026/Conference — Submitted to ICLR 2026_

### Official Review · Reviewer_jB8p · 2025-10-29

**Soundness:** 2
**Presentation:** 3
**Contribution:** 2
**Rating:** 4
**Confidence:** 4

**Summary:**

This paper proposes to attack code search-oriented large models in a black-box transfer manner. The adversary aims to increase the ranking of the desired code in the recommendation results of the model, given a natural language query. The adversary can add perturbations to identifiers to craft adversarial examples while maintaining the semantic consistency of the code. To optimize the transferability of the adversarial examples generated on the local surrogate model, the authors propose to approximately linearize the attack loss function to obtain the influence score of each token and then aggregate them to derive the influence score of the identifier. Experimental results show that the proposed attack technique exhibits stable transferability across different datasets and architectures, revealing the possibility of attacking large code models with a relatively small surrogate model.

**Strengths:**

1. The proposed transfer attack problem is very meaningful, especially for attacking large code models with restricted computational resources.
2. The mathematical and algorithmic techniques presented in this paper are elegant.

**Weaknesses:**

1. The linearity assumption may not hold for code models.
A minor perturbation in token space may induce a large divergence in the model feature space (e.g., "Desert" and "Dessert" differ by only a single token but diverge significantly in semantics), while a large perturbation in token space may induce only a small divergence in the model feature space (e.g., "Buy" and "Purchase" are very different in token space but may be very similar in semantic feature space). Thus, the fundamental assumption of this paper may not hold at all.
However, the proposed transfer attack heavily relies on the linear assumption:
(1) The first-order Taylor expansion requires the code model to be approximately linear;
(2) Aggregating the influence scores of tokens into the influence score of identifiers requires the code model to be approximately linear (Algorithm 1 in Appendix). Otherwise, simply summing the influence scores of individual tokens cannot accurately reflect the influence score of an identifier.
The reviewer suggests that the authors add two baseline attacks (random search and greedy search) to examine whether the linearity-based attack algorithm improves over them.


2. The attack perturbations are perceptible.
Appendix G-1.2 and G-2 show some clean code examples vs. adversarial code examples. For instance, the identifier "glm_ease_back_inout" in the clean code is replaced with "TemporaryBGCWigClassAttributeharitutdowntreat", which is excessively long. Given such lengthy perturbations, any type of attack can succeed very easily. The overly long perturbations can also be easily detected by humans and defensive detectors.

3. Missing related work
The following papers also discuss transfer attacks against code models:

Yang Y, Fan H, Lin C, et al. Exploiting the adversarial example vulnerability of transfer learning of source code. IEEE Transactions on Information Forensics and Security, 2024, 19: 5880-5894.

Liu Q, Ji S, Liu C, et al. A practical black-box attack on source code authorship identification classifiers. IEEE Transactions on Information Forensics and Security, 2021, 16: 3620-3633.

4. Missing baselines
The following papers are query-based black-box attacks. I suggest the authors also compare with them by attacking the surrogate model using these query-based attacks and then transferring the adversarial code to attack the target model to evaluate the transfer attack success rate.

Yang Z, Shi J, He J, et al. Natural attack for pre-trained models of code. Proceedings of the 44th International Conference on Software Engineering. 2022: 1482-1493.

Jha A, Reddy C K. Codeattack: Code-based adversarial attacks for pre-trained programming language models. Proceedings of the AAAI Conference on Artificial Intelligence. 2023, 37(12): 14892-14900.

Pu X, Xiong X, Li Y, et al. CodeSearchAttack: Enhancing soft-label black-box adversarial attacks on code. Journal of Information Security and Applications, 2025, 94: 104258.

Nguyen T D, Zhou Y, Le X B D, et al. Adversarial attacks on code models with discriminative graph patterns. arXiv preprint arXiv:2308.11161, 2023.

5. Missing threat model definition

6. Missing equation numbers

**Questions:**

1. About Tab. 6, it says, "In the black-box scenario, our attack transfer is more effective and requires just 0.01% of the API calls used by CodeAttack." Why does the black-box transfer attack still consume queries?

---

> ### Author Response · Authors · 2025-11-20
>
> We thank the reviewer for the detailed and constructive review. Below, we address your concerns:
>
> ## Linearity Assumption
> We would like to point out that the first-order Taylor expansion does not require the code model to be globally linear. As long as the distance between the modified and original input is reasonably small, the approximation holds. Furthermore, when aggregating the influences, the primary reason we can use a straightforward summation is the assumed linear independence between the different perturbation vectors $\delta_{i, t_x}$.
>
> Here are the experimental results for the random and greedy baselines on CosQA. The random baseline replaces identifier tokens with random tokens, while the greedy baseline uses the adversarial code after 5 greedy iterations on CodeT5+.
>
> | Attack Model | Eval Model | Sim Change | Pos Change Count | Pos Sim Change | Neg Change Count | Neg Sim Change |
> |:--- | :--- | :--- | :--- | :--- | :--- | :--- |
> | Random | CodeT5+ |  0.16 ± 6.06  | 5135 |  4.74 ± 3.74  | 4865 |  -4.67 ± 3.92  |
> | Random | OASIS |  1.44 ± 1.58  | 8478 |  1.86 ± 1.28  | 1522 |  -0.87 ± 1.02  |
> | Random | Nomic |  2.04 ± 3.55  | 7352 |  3.52 ± 2.64  | 2648 |  -2.08 ± 2.25  |
> | Random | voyage-code-3 |  1.51 ± 2.30  | 7636 |  2.38 ± 1.78  | 2364 |  -1.31 ± 1.33  |
> | Greedy (CodeT5+) | CodeT5+ |  22.19 ± 11.44  | 9687 |  22.99 ± 10.68  | 213 |  -4.04 ± 3.62  |
> | Greedy (CodeT5+)  | OASIS |  8.38 ± 5.31  | 9722 |  8.67 ± 5.11  | 245 |  -1.65 ± 1.69  |
> | Greedy (CodeT5+)  | Nomic |  13.84 ± 10.81  | 9105 |  15.55 ± 9.69  | 798 |  -3.99 ± 4.15  |
> | Greedy (CodeT5+)  | voyage-code-3 |  12.10 ± 8.08  | 9620 |  12.67 ± 7.69  | 380 |  -2.18 ± 2.54  |
>
> By comparing the random and greedy baselines with our method in Table 3 in our paper, our method exhibits both a higher positive similarity change ratio and a larger similarity change magnitude, demonstrating the effectiveness and validity of our current approach.
>
> ## Perceptible attack perturbations
> We acknowledge that the adversarially modified code snippets contain unusual or obfuscated variable names. However, we argue that renaming is also a widely adopted practice in industry for code obfuscation, typically employed to protect intellectual property or hinder reverse-engineering efforts [1, 2, 3, 4] by both the research community and commercial tools. Without domain expertise, it is difficult for users to distinguish between benign obfuscation and adversarial manipulation, making such attacks particularly insidious.
>
> Also, in many modern development workflows (e.g., AI-assisted "vibe coding"), retrieved results are often fed directly to LLMs and are not always validated by human developers. In such scenarios, as shown in Section 4.5, these adversarial snippets can degrade the quality and safety of the LLM's output.
>
>
> [1] Android Code Protection via Obfuscation Techniques: Past, Present and Future Directions
>
> [2] "Use Dotfuscator Community to Protect .NET Apps." Microsoft Learn, Microsoft, 12 Mar. 2025, learn.microsoft.com/en-us/visualstudio/ide/dotfuscator/?view=visualstudio.
>
> [3] PELock, https://www.pelock.com/products/pelock
>
> [4] Stunnix, http://stunnix.com/

---

> > ### Author Response · Authors · 2025-11-20
> >
> > ## Related Works
> >
> > We thank the reviewer for highlighting these related works. We acknowledge that the two papers on transferable attacks [5, 6] have evaluation settings similar to our work. However, we note several key distinctions:
> >
> > * Compared to **CodeTAE**[5], which focuses on the *code classification* task, our work targets the *NL-PL code search* task. This allows our attack to be tailored for more specific audiences, as discussed in our introduction.
> > * Compared to **SCAD**[6], our methodology does not require training a surrogate model for a transferable attack, thus it needs no knowledge of the target model's training data. Furthermore, unlike their rule-based approach, our gradient-based method is more general and can be applied to different programming languages, as demonstrated in Section 4.6.
> >
> > Regarding the other potential baselines:
> > * **CodeSearchAttack**[9] was published after our work's submission deadline.
> > * Designed for code-to-code tasks, **ALERT** [7] and **GraphCodeAttack** [10] require extracting patterns from input code, which is not feasible in our NL-to-PL code search context.
> > * We already include **CodeAttack**[8] as a black-box baseline in Section 4.3. Our results show that our method achieves both higher similarity improvement and a greater count of positive similarity changes.
> >
> >
> > [5]Yang Y, Fan H, Lin C, et al. Exploiting the adversarial example vulnerability of transfer learning of source code. IEEE Transactions on Information Forensics and Security, 2024, 19: 5880-5894.
> >
> > [6]Liu Q, Ji S, Liu C, et al. A practical black-box attack on source code authorship identification classifiers. IEEE Transactions on Information Forensics and Security, 2021, 16: 3620-3633.
> >
> > [7]Yang Z, Shi J, He J, et al. Natural attack for pre-trained models of code. Proceedings of the 44th International Conference on Software Engineering. 2022: 1482-1493.
> >
> > [8]Jha A, Reddy C K. Codeattack: Code-based adversarial attacks for pre-trained programming language models. Proceedings of the AAAI Conference on Artificial Intelligence. 2023, 37(12): 14892-14900.
> >
> > [9]Pu X, Xiong X, Li Y, et al. CodeSearchAttack: Enhancing soft-label black-box adversarial attacks on code. Journal of Information Security and Applications, 2025, 94: 104258.
> >
> > [10]Nguyen T D, Zhou Y, Le X B D, et al. Adversarial attacks on code models with discriminative graph patterns. arXiv preprint arXiv:2308.11161, 2023.
> >
> >
> > ## Threat Model:
> > We thank the reviewer for pointing out the necessity of the threat model, and would like to clarify it as follows. We will update and clarify this section in the revised draft.
> >
> > ### Adversary’s Capability
> >
> > For a given query, the adversary can inject several modified code snippets (irrelevant or malicious) into the retrieval corpus. The adversary can perform token-level perturbations on these snippets, provided the perturbations maintain the snippet's functionality (i.e., preserve the AST structure).
> >
> > ### Adversary’s Knowledge:
> >
> > Given a query and a code snippet, the adversary has white-box access to a single, known code embedding model (e.g., CodeT5+). Using this model, the adversary can calculate two things: (1) the similarity between the query and the code snippet, and (2) the gradient of this similarity with respect to each token's embedding in the code snippet.
> >
> > ### Adversary’s Goal:
> >
> > Given a query Q and an irrelevant or malicious code snippet C, the adversary’s goal is to generate an adversarial code snippet C’ such that the similarity between $(Q, C') \geq \tau$. Here, $\tau$ is a dynamic threshold sufficient to ensure C' is ranked in the top-k retrieval results.

---

> ### Comment · Reviewer_jB8p · 2025-11-25
> **Response**
>
> The authors' response makes sense to me. However, given that there are already many studies on large code models, as acknowledged by the authors, shifting the task to code retrieval does not make substantial contributions. The proposed method is essentially a common feature-based adversarial attack, which can also be applied to any code task, but does not specifically work for retrieval. The authors should compare their method to others in other code tasks.

---

### Official Review · Reviewer_XsUi · 2025-10-31

**Soundness:** 2
**Presentation:** 3
**Contribution:** 2
**Rating:** 4
**Confidence:** 4

**Summary:**

This paper introduces a language-agnostic transferable adversarial attack method that exploits the vulnerability of CLMs, which perturbs (renames) identifier tokens in code snippets (without changing functionality) to make an irrelevant snippet appear highly relevant to the target query.
Experiments demonstrate that adversarial code snippets generated using smaller models are highly transferable to larger or closed-source models, thus significantly reducing the accuracy of mainstream models in code retrieval tasks.
Furthermore, it reveals that current state-of-the-art code search models rely on lexical features rather than deeper code understanding, exposing significant robustness gaps despite high benchmark performance and highlighting the need for future improvement.

**Strengths:**

1. Well-defined and important problem: the paper reveals a critical vulnerability previously overlooked in code search models.

2. Strong cross-model transferability: adversarial examples crafted on small models effectively fool large and even closed-source models, with >95% success rates.

3. Detailed evaluation: experiments cover multiple models, datasets, and five programming languages with detailed analysis.

**Weaknesses:**

1. Lack of methodological novelty: the core attack adapts the well-known gradient-based token substitution without major algorithmic novelty (Q1).

2. Lack of rigorous theoretical analysis on attack transferability: there is no formal proof on the transferability of adversarial attacks, which leaves the generalizability and robustness of this work in question (Q2-Q3).

3. Limited quantification on attack margins: identifier renaming may introduce unnatural names, potentially making adversarial code detectable (Q4).

4. Insufficient experimental validation: the evaluation lacks a comparison with random attacks as well as a more comprehensive assessment of the impact on downstream tasks, which are necessary to validate the effectiveness and generalizability of the attack (Q5).

5. Some minor writing problems.

**Questions:**

1. Given that the proposed approach aims to heuristically increase the similarity between a code snippet and a target query, while leaving the initial embedding space unchanged, I am wondering whether the constraints of the embedding space or vocabulary introduce inherent limitations. For instance, in the example outlined in Section 1, where an attacker aims to push malicious code to users with high computational resources, if the malicious code segment does not exist within the model's embedding space, how can this method be effectively deployed?

2. The authors defer the investigation into the causes of the strong correlation in attack transferability across different models to future work. However, I believe that an in-depth theoretical analysis and discussion of this correlation, supported by controlled ablative studies, should constitute a key contribution of this paper. For instance, comparing the transferability performance on models pre-trained on distinctly different corpora but with identical architectures, or vice versa, would be highly informative. Even experiments involving locally-trained, relatively simple models could substantiate the conclusions.

3. While the attacks crafted with a small model are mostly effective on larger ones, one case (attacking with OASIS and evaluating on CodeT5+) showed lower success (e.g. ~82% precision on CosQA). Could the authors elaborate on conditions where the transferability does not hold or be less robust? For example, if the target model uses a drastically different embedding space or tokenization, would the attack still work?

4. How realistic is it for an attacker to introduce adversarial code without being noticed? The attack alters identifier names in code, which could introduce odd or semantically meaningless names to increase similarity with the target query. In practice, a code snippet with bizarre variable names (chosen to match a query’s keywords) might alert a vigilant developer or be flagged by simple static analysis.

5. How to prove that the performance change observed under the proposed method is different from the baseline that merely makes random perturbations in the embedding space? For example, in the Experiments RQ4 of CodeAttack [1], the authors compared CodeAttack against a variant, CodeAttackRAND, which randomly samples tokens from the input code for substitution. Similarly, a comparative experiment involving such random attacks is expected.

[1] Jha, A., & Reddy, C. K. (2023). CodeAttack: Code-Based Adversarial Attacks for Pre-trained Programming Language Models. Proceedings of the AAAI Conference on Artificial Intelligence, 37(12), 14892-14900.

6. There are some writing and formatting errors in the paper, for example:
   - In Introduction, "closed-sourced (Voyage-code-3)" –> "closed-source (Voyage-code-3)".
   - In Introduction, "for example pushing malicious cryptomining code..." –> "for example, pushing malicious cryptomining code...".
   - In the title of Table 5 "...that the attack transfer effects, while present, is less predictable...", "is"->"are"

---

> ### Author Response · Authors · 2025-11-20
>
> We thank the reviewer for the detailed and constructive review. Below, we address your concerns:
>
> ## Our Contribution
> We acknowledge that our methodology is similar to gradient-based methods used in previous code classification tasks. Our contribution lies in how this mechanism is instantiated for NL→code retrieval and transfer attacks, namely:
> * We optimize similarity with natural-language queries and evaluate impact on ranking metrics and real-world ranking corruption, not just embedding distances or classification labels.
>
> * We systematically show cross-model transfer from a small model to larger and closed systems, under realistic corpora and multiple languages.
>
> * We also validate the attack in a realistic corpus-injection scenario with actual malware, as demonstrated in Section 4.4 and the later new real-world experiment.
>
> ### Adding malicious code to the embedding space
> According to [1], the adversarially modified code snippets can be injected into the embedding space by uploading them to multiple platforms, such as StackOverflow or GitHub. For example, the StackOverflow dataset (https://www.kaggle.com/datasets/stackoverflow/stackoverflow) is updated quarterly. This provides an opportunity for the vector representing malicious code snippets to be indexed and added to the embedding space over time.
>
> [1] You see what I want you to see: poisoning vulnerabilities in neural code search
>
>
>
> ## Regarding Transferability
> We thank the reviewer for pointing out the lower precision when transferring from OASIS to CodeT5+. OASIS is trained on robustness-related tasks, and we hypothesize that this adds special features to the embedding space of OASIS, making attacks generated on OASIS less transferable to CodeT5+.
>
> Meanwhile, we believe that differences in tokenization or internal model dimensions do not significantly influence transferability. As mentioned in Table 1 in our paper, the models use different tokenizers and thus have different vocabularies. Furthermore, the internal dimension of CodeT5+ and the other three models are not the same, yet attacks still transfer well from CodeT5+ to the other models.

---

> > ### Author Response · Authors · 2025-11-20
> >
> > ## Practicality of the Attack
> >
> > We acknowledge that the adversarial attack introduces unusual variable names into the code snippets. However, in many RAG systems, the retrieved results are viewed only by the LLM (which is less sensitive to such artifacts) rather than by human developers. Additionally, the rise of "vibe coding" increases the likelihood that the retrieved code snippets are never validated by humans. In these cases, these adversarially modified code snippets already degrade the performance of RAG systems, as shown in Section 4.5.
> >
> > To illustrate the practical relevance of this challenge, consider a scenario in which a novice developer queries for a solution to a standard programming task. If an attacker successfully elevates a malicious script to the top of the retrieval results, the developer is likely to execute this "top-ranked" result.
> >
> > ### Real-world experiment
> > To empirically demonstrate the severity of this threat, we designed a real-world experiment simulating an attack on novice developers:
> >
> > Query Set: We sampled 20 novice-level software engineering questions from StackOverflow to mimic the search behavior of less experienced developers who rely heavily on search rankings.
> >
> > Code Corpus: We constructed a realistic retrieval database combining benign sources (accepted answers to the sampled questions and 2,000 samples from a public StackOverflow Python dataset [1]) with a Malicious Set of 33 Python programs, including known malware and CVE proof-of-concepts.
> >
> > We applied our proposed attack to the malicious scripts and measured their retrieval rankings. As shown in Table 1, prior to the attack, the models effectively filtered out malicious code (near 0 at Top-10). However, the attack successfully pushed malicious samples into prominent positions. On average, a developer reviewing the top 10 results would be exposed to approximately 2.6 and 1.8 malicious scripts for CodeT5+ and OASIS, respectively. This confirms that our method can effectively bypass the modern code search models and expose users to severe security risks.
> >
> >
> > **Table 1: Average count of malicious code snippets appearing in top-k results before and after the attack.**
> >
> > | Exp_Name | Count@1 | Count@5 | Count@10 |
> > | :--- | :--- | :--- | :--- |
> > | CodeT5+ Initial | 0.00 | 0.10 | 0.15 |
> > | OASIS Initial | 0.00 | 0.00 | 0.05 |
> > | CodeT5+ Attacked | **0.25** | **1.70** | **2.60** |
> > | OASIS Attacked| **0.05** | **0.85** | **1.80** |
> >
> > [1] https://huggingface.co/datasets/suriyagunasekar/stackoverflow-python-with-meta-data
> >
> >
> >
> > ## Random Baseline
> > We agree that comparing against random perturbations is important. We have conducted experiments on random token replacement, but we omitted them from the main paper as their performance was too poor to serve as a meaningful baseline. We provide the performance of the random baseline below for the reviewer's reference. These results will be added to the revised version.
> >
> > | Dataset | Attack Model | Eval Model | Sim Change | Pos Change Count | Pos Sim Change | Neg Change Count | Neg Sim Change |
> > | :--- | :--- | :--- | :--- | :--- | :--- | :--- | :--- |
> > | CosQA | Random | CodeT5+ | 0.16 ± 6.06 | 5135 | 4.74 ± 3.74 | 4865 | -4.67 ± 3.92 |
> > | | Random | OASIS | 1.44 ± 1.58 | 8478 | 1.86 ± 1.28 | 1522 | -0.87 ± 1.02 |
> > | | Random | Nomic | 2.04 ± 3.55 | 7352 | 3.52 ± 2.64 | 2648 | -2.08 ± 2.25 |
> > | | Random | voyage-code-3 | 1.51 ± 2.30 | 7636 | 2.38 ± 1.78 | 2364 | -1.31 ± 1.33 |
> > | CLARC | Random | CodeT5+ | 0.36 ± 4.38 | 5403 | 3.43 ± 2.87 | 4597 | -3.24 ± 2.82 |
> > | | Random | OASIS | 1.17 ± 1.79 | 7861 | 1.78 ± 1.36 | 2139 | -1.10 ± 1.30 |
> > | | Random | Nomic | 1.11 ± 3.32 | 6382 | 2.91 ± 2.40 | 3618 | -2.07 ± 2.10 |
> > | | Random | voyage-code-3 | 0.68 ± 2.17 | 6172 | 1.92 ± 1.59 | 3828 | -1.34 ± 1.30 |
> >
> > As presented in the table, the baseline of randomly switching tokens generates slightly more positive similarity changes than negative ones. However, the magnitude of these positive and negative changes is very close. In contrast, our gradient-based method is significantly more effective. In most cases, our method achieves positive changes in over 90% of the evaluated (query, code snippet) pairs on the attack model and via transferability, much higher than the positive change ratio of the random baseline. Furthermore, the magnitude of the similarity change from our method is typically greater than 10%, whereas the change in the random baseline remains below 3%.
> >
> > ## Presentation Issues:
> > We apologize for the grammatical and formatting errors. We will thoroughly proofread the entire paper and fix these issues in the revised version.

---

### Official Review · Reviewer_dLaj · 2025-11-01

**Soundness:** 2
**Presentation:** 3
**Contribution:** 2
**Rating:** 4
**Confidence:** 4

**Summary:**

This paper proposes an adversarial attack method for code search, which perturbs the code snippet to maximize its similarity with the target query while preserving functionality. The authors show that changes made using smaller models, such as CodeT5+, are highly transferable to larger or closed-source models like Nomic-emb-code or Voyage-code-3. These changes can increase the similarity between a query and an irrelevant code snippet, reducing key retrieval metrics such as Mean Reciprocal Rank (MRR) of state-of-the-art models by up to 40%. Experimental results show the vulnerability of current code search methods and the need for more robust, semantic-aware approaches.

**Strengths:**

- This work presents a dedicated adversarial attack against code search models, identifying a significant threat vector.

- The paper demonstrates the vulnerability of current code search methods, as attacks transfer effectively across model sizes and architectures, including closed-source systems.

**Weaknesses:**

1. Lack of Demonstrated Practical Impact: The threat model is narrowly defined. The paper relies on embedding similarity as a proxy for success, failing to demonstrate consequential failures in downstream tasks (e.g., corrupting code retrieval or evading a vulnerability scanner). The threat model should be refined and grounded in practical scenarios to establish true utility and severity.

2. Limited Evaluation Setup: The attack is highly targeted, requiring a specific code snippet as an objective. This overlooks the more general and practical threat of non-targeted attacks, which aim to cause arbitrary misclassification or degradation, and should be evaluated to assess the full scope of the vulnerability.

3. Insufficient Comparisons: The chosen baseline (Code Attack) is designed for classification, not retrieval. A comparison against state-of-the-art adversarial retrieval methods is missing, undermining the claim of effectiveness. Furthermore, there are many recent code LLMs and code search models, which the paper fail to evaluate.

Previously work, such as (Zhang et al., 2023) and (Tian et al., 2023), already described semantic-preserving code transformations for evaluating machine learning-based code models. The paper can discuss and compare.

W. Zhang, et al.,  "Challenging Machine Learning-Based Clone Detectors via Semantic-Preserving Code Transformations," in IEEE Transactions on Software Engineering, vol. 49, no. 5, pp. 3052-3070, 1 May 2023.

Tian Z, Chen J, Jin Z. Code difference guided adversarial example generation for deep code models, ASE 2023.

4. Narrow Evaluation Focus: The assessment relies solely on ranking metrics (e.g., MRR and Recall), failing to evaluate the semantic correctness or potential maliciousness of the retrieved adversarial code, which is critical for assessing real-world impact.

5. This paper introduces a scenario where ~10% of irrelevant snippets in a small candidate pool are adversarially edited, but the evaluations rely on offline pools (e.g., ~500 items in CosQA) and sampled pairs. The setup shows sensitivity under controlled conditions, but it does not establish exploitability in real repositories with large-scale indexing, deduplication, and ingestion pipelines.

6. This paper claims to preserve functionality by ensuring the consistency of the AST (Abstract Syntax Tree) structure. However, this cannot fully guarantee semantic consistency. The reason is that identifier renaming may lead to naming conflicts or break code that relies on reflection and specific naming conventions—for instance, code that requires specific identifier names to interact with external APIs.

7. Presentation Issues: The paper suffers from some presentation issues, such as  lack of figures, and grammatical errors.

**Questions:**

Please refer to “Weaknesses”.

---

> ### Author Response · Authors · 2025-11-20
>
> We thank the reviewer for the detailed and constructive review. Below, we address your concerns:
>
> ## Practical Impact and Additional Evaluation
>
> To demonstrate the practical impact of our approach beyond targeted attacks, we evaluated the method in a realistic attack scenario. We define a threat model in which a novice developer queries for solutions to standard programming tasks. In this context, the successful elevation of a malicious script to a top-ranked position significantly increases the likelihood that the developer will execute the compromised code.
>
> ### Experimental Setup
> - **Query Set:** We collected 20 generic queries corresponding to common software engineering tasks to emulate the search behavior of novice developers.
> - **Mixed Corpus:** The retrieval corpus combines benign sources (verified accepted answers and 2,000 samples from a public StackOverflow dataset [1]) with a **Malicious Set** of 33 executable Python scripts, including known malware and CVE proof-of-concepts.
>
> We measured the attack's efficacy in degrading search engine reliability using Count@K metrics, which represent the average number of malicious code snippets in the top-k retrieval results. As shown in Table 1, without the attack, the malicious scripts typically rank outside the top 10. Post-attack, however, malicious samples consistently appear within the top-10 rankings. Specifically, an average of 1.70 (CodeT5+) and 0.85 (OASIS) malicious snippets appeared in the top-5 results.
>
> These results demonstrate that the threat model constitutes a practical, generalized retrieval attack, confirming the adversary's ability to inject specific code snippets into top-ranked search results.
>
> **Table 1: Evaluation of Generalized Ranking Corruption. "Count@K" denotes the average number of malicious scripts displacing benign results in the top-K rankings.**
>
> | Exp_Name | Count@1 | Count@5 | Count@10 |
> | :--- | :--- | :--- | :--- |
> | CodeT5+ (Clean) | 0.00 | 0.10 | 0.15 |
> | OASIS (Clean) | 0.00 | 0.00 | 0.05 |
> | **CodeT5+ (Attacked)** | **0.25** | **1.70** | **2.60** |
> | **OASIS (Attacked)**| **0.05** | **0.85** | **1.80** |
>
> [1] https://huggingface.co/datasets/suriyagunasekar/stackoverflow-python-with-meta-data
>
>
>
>
> ## Comparison with other work
> We thank the reviewer for highlighting these two relevant papers on semantic-preserving code transformation. We will update our related work section to explicitly compare our method against them and clarify the novel contributions of our approach.
>
> To our knowledge, our work is among the first to address transferable adversarial retrieval in the natural language-to-code setting, while the two works mentioned by the reviewer both focus on code-to-code attacks. DRLSG[2] aims to decrease similarity between code snippets. In contrast, our NL-to-code method aims to increase the similarity between a specific user query and an arbitrary code snippet, allowing our method to target specific queries, a capability we demonstrated in our experiments. Methodologically, CODA[3] employs a rule-based attack that requires sampling multiple code snippets, whereas our gradient-based method is more generalizable and requires only a single target query for optimization.
>
> [2] W. Zhang, et al., "Challenging Machine Learning-Based Clone Detectors via Semantic-Preserving Code Transformations," in IEEE Transactions on Software Engineering, vol. 49, no. 5, pp. 3052-3070, 1 May 2023.
>
> [3] Tian Z, Chen J, Jin Z. Code difference guided adversarial example generation for deep code models, ASE 2023.
>
> ## Regarding the semantic correctness and maliciousness of the retrieved adversarial code
> Our methodology is designed to preserve semantic correctness. We ensure this by preserving the Abstract Syntax Tree (AST) of the code snippet, verifying the AST after each transformation. Moreover, we only modify parser-found identifiers that are **defined within the scope** of the current code snippet. This approach prevents unintended changes to arguments for external API calls and avoids potential naming conflicts. As a result, we are confident that the generated adversarial code snippets remain semantically correct and compilable.
>
> We acknowledge the reviewer's point that our current experiments use benign (though irrelevant) code snippets. As detailed in our new experiment, we address this limitation by using real-world malicious code as the foundation for our adversarial examples.
>
>
>
> ## Presentation Issues
> We sincerely apologize for the grammatical errors. We will thoroughly proofread the entire draft. We will also add a figure to better illustrate our methodology in the revised version, as suggested.

---

### Official Review · Reviewer_CwLQ · 2025-11-01

**Soundness:** 3
**Presentation:** 3
**Contribution:** 2
**Rating:** 4
**Confidence:** 4

**Summary:**

This paper investigates the robustness of neural code language models (CLMs) used for code retrieval against adversarial attacks. The authors propose a language-agnostic, transferable adversarial attack that perturbs non-functional code elements (mainly identifiers) to mislead code search models without changing the program’s semantics. They demonstrate that adversarial examples generated using smaller open-source models such as CodeT5+ can successfully transfer to larger or closed-source models, including Nomic-emb-code and Voyage-code-3. Experimental results show that such perturbations can degrade retrieval performance—reducing MRR by up to 40%—thus revealing significant vulnerabilities in current CLM-based code search systems.

**Strengths:**

+ The experimental evaluation is comprehensive. The authors conduct extensive experiments on both CodeT and OASIS, demonstrating the consistency and robustness of their findings across different code search models.
+ The transferability results are especially interesting — adversarial examples produced with small models effectively deceiving large or closed-source systems is a convincing and practically relevant finding.

**Weaknesses:**

- Incremental contribution. The authors claim “We propose one of the first adversarial attack methods for code search”; however, prior work (e.g., [1,2]) has already explored adversarial attacks on code retrieval systems. Although the authors argue that previous studies mainly targeted classification tasks, this distinction is not convincing, as those works share similar attack mechanisms and objectives. Consequently, the novelty of the proposed approach appears rather incremental.

- Weak motivation. The motivation is weak and somewhat overstated. Although the authors claim that adversarial attacks on code search present “unique challenges”, the differences from classification tasks are superficial. The example of “targeted manipulation of search results” is speculative and lacks empirical support, and the reliance on “offline embeddings” does not constitute a fundamentally new threat model, as similar vulnerabilities exist in other embedding-based systems. I suggest the authors provide a detailed, realistic example to better illustrate the practical relevance of their claimed challenges.

- Unclear Method. The authors seem to have placed much of the theoretical or preliminary content within the method section (e.g., Section 3.1 Gradient-Based Method), which mainly contains derivations rather than concrete methodological descriptions. I suggest adding an overall schematic figure to clearly illustrate the attack pipeline — including the inputs, outputs, and the overall process of how the attack is performed.


[1] Codeattack: Code-based adversarial attacks for pre-trained programming language models

[2] You see what I want you to see: poisoning vulnerabilities in neural code search

**Questions:**

1. Could the authors clarify how their proposed attack method fundamentally differs from prior adversarial attacks on code retrieval or classification models?

2. Can the authors offer a concrete, realistic scenario or case study demonstrating the practical impact of such attacks in real-world code search systems?

---

> ### Author Response · Authors · 2025-11-20
>
> We thank the reviewer for the detailed review. Below we address your concerns:
>
>
> ## Our Contribution and Difference from Prior Work
>
> We acknowledge that our attack builds on gradient-based methods, which have been explored in code classification settings. However, our contribution is to systematically adapt and specialize this family of attacks to the code search scenario, which introduces several distinct aspects:
> * **Retrieval-specific objective:** Instead of flipping a single label, our attack directly optimizes the query–code similarity that drives ranking, enabling the adversary to push arbitrary code (including malicious ones) into the top results for natural-language queries.
> * **Black-box transfer to large/closed models:** We show that adversarial examples crafted on a small, open-source model reliably transfer to larger and closed-source models across multiple programming languages. To the best of our knowledge, prior work has not demonstrated such cross-architecture transfer in the NL->code retrieval setting.
> * **Realistic, end-to-end ranking manipulation:** We also perform a real-world study in which our attack lifts known malicious Python scripts into the top-5 or top-10 results for queries from novice programmers, demonstrating end-to-end exploitability of code search systems.
>
>
> We will include a figure in the paper to better illustrate the pipeline of our transfer attack, thereby enhancing the clarity of our methodology. Finding the adversarial code snippet is a three-step process: First, we select a small code embedding model (the "Attack Model"), such as CodeT5+, and use it to embed the target query and the code snippet. Second, we use these embeddings to calculate their similarity and then backpropagate the gradient to the code snippet's token embeddings. Lastly, we use this gradient to calculate the “influence” of replacing tokens in identifiers with other valid vocabulary tokens. We then generate the adversarial code by replacing tokens based on this calculated influence.
>
> In our experiments, we iterate this process and select the code snippet with the highest similarity to the query as the final adversarial code. The empirical results show that these adversarial snippets also demonstrate improved similarity to the query when tested on other, larger or black-box code search models.

---

> > ### Author Response · Authors · 2025-11-20
> >
> > ## Practical Impact and Threat Validation
> >
> > To address concerns regarding the practical relevance of ranking manipulation in code search, we clarify the distinction between this threat model and standard classification attacks. In classification, an adversarial success results in a label error; in code search, the consequence is the potential **direct execution of compromised code**. To illustrate the practical relevance of this challenge, consider a scenario in which a novice developer queries for a solution to a standard programming task. If an attacker successfully elevates a malicious script to the top of the retrieval results, the developer is likely to execute this "top-ranked" result.
> >
> > ### Real-world experiment
> >
> > We designed a real-world experiment using authentic user queries and verifiable malicious code:
> >
> > - **Query Set:** We sampled 20 novice-level software engineering questions from StackOverflow to mimic the search behavior of less experienced developers.
> > - **Code Corpus:** Our retrieval database combined benign sources (code snippets from the accepted answers to the 20 questions and 2,000 samples from a public StackOverflow Python dataset [1]) with a **Malicious Set** of 33 Python programs comprising known malware and CVE proof-of-concepts.
> >
> > For each query, we applied our proposed attack to modify the malicious scripts. As illustrated in Table 1, prior to the attack, malicious code rarely breached the top rankings (near 0 at Top-10). However, the attack successfully forced malicious samples into prominent positions. On average, a developer reviewing the top 10 results would encounter approximately 2.6 and 1.8 malicious scripts for CodeT5+ and OASIS, respectively. This demonstrates that the threat is not merely theoretical; our method effectively bypasses the semantic filters of state-of-the-art models, exposing developers—particularly novices who rely heavily on top-ranked results—to severe security risks.
> >
> > **Table 1: Average count of malicious code snippets appearing in top-k results before and after the attack.**
> >
> > | Exp_Name | Count@1 | Count@5 | Count@10 |
> > | :--- | :--- | :--- | :--- |
> > | CodeT5+ Initial | 0.00 | 0.10 | 0.15 |
> > | OASIS Initial | 0.00 | 0.00 | 0.05 |
> > | CodeT5+ Attacked | **0.25** | **1.70** | **2.60** |
> > | OASIS Attacked| **0.05** | **0.85** | **1.80** |
> >
> > [1] https://huggingface.co/datasets/suriyagunasekar/stackoverflow-python-with-meta-data
> >
> >
> >
> >
> > ## Differences from Previous Works
> > We thank the reviewer for mentioning the two relevant previous works. We would like to point out that the method from CodeAttack[2] modifies the functionality of the original code snippet and thus falls outside of our threat model. Meanwhile, "You see what I want you to see" [3] aims to inject poisonous data into the training set of a CLM. This is distinct from our attack model, which targets the retrieval corpus at inference time and does not assume access to the training process.
> >
> > [2] Codeattack: Code-based adversarial attacks for pre-trained programming language models
> >
> > [3] You see what I want you to see: poisoning vulnerabilities in neural code search

---

### Meta-Review · Area_Chair_c51i · 2025-12-24

**Summary:**

This paper presents a new adversarial attack against code language models with the features of high transferability and language-agnostic. By perturbing identifies within the code snippet, the attacker can possibly mislead the model to search for the malicious code.

All the four reviewers gave the rating of 4. The authors have provided detailed responses to their comments. AC has read through the paper, all the reviews, and responses, and believes that the critical concerns are still not fully addressed, as elaborated below.

1.	All the reviewers pointed out the issue of incremental contributions. They also listed some relevant works, asking the authors to compare. The authors have presented detailed clarification and comparisons. However, AC still thinks the technical contributions are insufficient. Even this is the first attack that targets the code search scenario, but technically it still uses the conventional gradient-based approach. It just adapts the optimization objective to the new scenario, which is not significant. The authors also claimed another contribution is the transferability. But achieving this is very simple and conventional: just using one small model to generate patch and attack another one. This is also not a big contribution.
2.	All the reviewers also mentioned about the practical significance, and real-world impact. We appreciated that the authors provided new experiments to demonstrate the real-world impact. However, AC thinks this cannot fully meet the reviewers’ expectations. It only shows the attacker’s code can be searched, but not the subsequent consequences, like how it will compromise the entire code project in the real world, whether it can bypass existing scanners, etc.
3.	The authors also overlooked some other comments in their rebuttal, such as narrow evaluation focus and real-world set up (dLaj), lacking theoretical analysis about transferability (XsUi), etc.

AC believes the above concerns are critical and cannot be addressed. Hence, this paper is recommended as rejection.

**Reviewer Concerns:**

Some comments about technical detail clarification can be addressed, such as "semantic correctness and maliciousness of the retrieved adversarial code", new experiments of Random Baseline, Linearity Assumption and Perceptible attack perturbations. But some criical comments are still outstanding, as described above, including limited technical contribution, practical impact, etc.

**Reviewer Scores:**

I think the reviewers will not adjust their scores, as the outstanding critical comments are shared by all of them.

---

### Decision · Program_Chairs · 2026-01-26

Reject